# Space-Air-Ground Integrated 6G Wireless Communication Networks: A Review of Antenna Technologies and Application Scenarios

**DOI:** 10.3390/s22093136

**Published:** 2022-04-20

**Authors:** Francesco Alessio Dicandia, Nelson J. G. Fonseca, Manlio Bacco, Sara Mugnaini, Simone Genovesi

**Affiliations:** 1IDS Ingegneria dei Sistemi S.p.A., 56121 Pisa, Italy; 2Antenna and Sub-Millimetre Waves Section, European Space Agency (ESA), 2200 AG Noordwijk, The Netherlands; nelson.fonseca@esa.int; 3Institute of Information Science and Technologies (ISTI), National Research Council (CNR), 56124 Pisa, Italy; manlio.bacco@isti.cnr.it; 4OneWeb, London W12 7FQ, UK; smugnaini@oneweb.net; 5Dipartimento di Ingegneria dell’Informazione, University of Pisa, 56122 Pisa, Italy; simone.genovesi@unipi.it

**Keywords:** space-air-ground communication network, 5G, 6G, millimeter waves, massive MIMO, CubeSat, satellite internet access, Internet of Things, UAV, LAP, HAP, antenna, phased array

## Abstract

A review of technological solutions and advances in the framework of a Vertical Heterogeneous Network (VHetNet) integrating satellite, airborne and terrestrial networks is presented. The disruptive features and challenges offered by a fruitful cooperation among these segments within a ubiquitous and seamless wireless connectivity are described. The available technologies and the key research directions for achieving global wireless coverage by considering all these layers are thoroughly discussed. Emphasis is placed on the available antenna systems in satellite, airborne and ground layers by highlighting strengths and weakness and by providing some interesting trends in research. A summary of the most suitable applicative scenarios for future 6G wireless communications are finally illustrated.

## 1. Introduction

The disruptive growth of the wireless communication system performance requirements, such as data throughput, energy efficiency, latency and security, along with the Internet of Things (IoT) [1,2] paradigm are stimulating the research and development for novel solutions to serve the highest possible number of users and manage sensor networks with the required degree of flexibility and scalability. In the past, the exploitation of larger frequency bandwidths and the network densification, namely the deployment of more and more Base Stations (BSs) to reduce the cell area, were adopted to tackle the ever-increasing data throughput demand. Conversely, in the upcoming fifth generation (5G) wireless communication systems technology, the Spectral Efficiency (SE) improvement is assured primarily by the massive Multiple-Input–Multiple-Output (MIMO) technology [3,4,5]. Specifically, massive MIMO systems rely on the Space Division Multiple Access (SDMA) technique to achieve a multiplexing gain by serving multiple users simultaneously with the same time-frequency resource [6,7,8,9]. Its implementation is based on BSs equipped with Active Electronically Steerable Antenna (AESA) arrays [10] composed of a massive number of radiating elements in order to provide advanced beamforming methods [11,12,13,14] capable of sending different streams of data allocated on the same time-frequency resource to different users within the cell [15,16].

The deployment of 5G wireless communication infrastructure started in various parts of the world around 2019 [17]. While network installation and distribution are underway, researchers started to investigate next sixth generation (6G) wireless communication networks [17,18,19]. The ubiquitous and seamless wireless connectivity, one of the many 5G goals, cannot be satisfied by exploiting only terrestrial infostructures. Terrestrial BSs cannot be deployed in off-grid or inaccessible areas such as rural zones, deserts, oceans and, in general, in harsh and remote environments. Thereby, the integration of Unmanned Aerial Vehicle (UAV)-assisted wireless communications into 5G systems has attracted tremendous interest in the last few years [20,21,22,23,24,25].

Although the research on 6G is still in its infant stage [26], it is envisioned that the concept of anytime and anywhere network access undergoes breakthroughs with the advent of next wireless communication generation with the fruitful integration of space, air and ground networks in the framework of a Vertical Heterogeneous Network (VHetNet) [27,28]. To this end, it will be necessary to consider, and to manage, the coexistence of different wireless connectivity platforms from ground segment to space segment composed of dissimilar software and hardware architectures, network topologies and communications protocols. Artificial Intelligent (AI) and Machine learning (ML) technology will play an increasingly crucial role within network management and automation and to meet the reconfigurability demand [29]. Figure 1 shows a sketch of a VHetNet scenario by considering some satellite, airborne and terrestrial communication networks, vital features for ubiquitous and seamless purposes. As schematically shown, the overall network comprises three main layers: space, air and ground segments. While both the terrestrial and space segments are well-established telecommunication connectivity services, they face a variety of respective drawbacks and challenges. Thereby, to solve or partially mitigate these problems, the air communication layer will play an important complementary role for future wireless communication systems in providing universal and favorable access to the global network with the required Quality of Services (QoS) [30].

In general, the air segment turns out to be essentially based on UAVs, also known as drones or atmospheric satellites, especially for wireless communication missions. Owing to their autonomy, flexibility, versatility, and of contained CAPital EXpenditure (CAPEX) and OPerating EXpenditure (OPEX), UAVs are becoming a more and more appealing option [31,32]. However, in general, depending on the mission applications and goals, these flying platforms may be manned as well [33].

In addition to the network topologies and architectures, the exploitation of large frequency spectrum is pivotal for supporting communication links with adequate QoS and deal with the ever-increasing wireless communications system’s needs. Therefore, besides the sub-6 GHz frequency bands, the millimeter-wave (mmWave) spectrum, namely frequencies in the range of 30–300 GHz, will be promising for the next wireless communications systems. For this reason, the mmWave band has recently drawn great attention for 5G and beyond wireless communications systems [34,35,36,37] to support higher data rates due to greater bandwidth.

However, despite the advantage of a large spectrum, mmWave signal propagations are prone to some impairments with respect to those in the sub-6 GHz range [38,39]. Significant propagation loss, lower coherence time due to rapid channel fluctuation, superior power consumption in the analog-to-digital (A/D) conversion, higher sensitivity to radio-wave blockage and a low power amplifier efficiency represent some of the challenges that mmWave communications have to tackle [40,41,42].

Moreover, looking forward to the 6G era and beyond, the exploitation of even higher carrier frequencies such as terahertz (THz) or optical frequency bands are envisioned to play a crucial position by providing high bandwidth and huge components miniaturization [43]. Nevertheless, THz or optical communications reach out to stronger hardware challenges including antennas, power amplifiers, or modulators [44].

This article provides a general overview concerning Space-Air-Ground Integrated Network (SAGIN) and emphasizes some research activities to support the multi-dimensional and interoperational network of the future 6G wireless communications and beyond. Specifically, particular attention has been addressed on the available antenna systems in satellite, airborne and ground layers by highlighting strengths and challenges as well as by providing some promising research directions. Antennas are certainly among the most fundamental components, and they are determinant for the performance of the onboard transceiver subsystem. Table 1 provides a comparison among the review papers published in recent years dealing with 6G and SAGIN technology, highlighting the particular angle from which they have looked to this broad topic.

This paper is organized as follows. Section 2 discusses the space segment and developments, including a particular focus on satellite constellations, followed by a thorough overview on antenna technologies currently used onboard advanced satellite systems and under development for future satellite systems. A comprehensive investigation on Low Altitude Platform (LAP) and High Altitude Platform (HAP) challenges such as network topology, Spectral Efficiency (SE) and antennas technologies is reported in Section 3, whereas the ground segment is introduced in Section 4. Section 5 is devoted to the examination of the various application scenarios and potential opportunities regarding the paradigm of SAGIN in the future 6G wireless communications. Finally, the conclusions are reported in Section 6.

## 2. Space Segment

From the modest radio transmitter onboard Sputnik 1 in the late 1950s to currently developed Very High Throughput Satellite (VHTS) systems, there has been a great deal of space technology developments and innovations, driven by new applications with communication satellites at the forefront of the commercial use of space. The turn of the century marked a major paradigm shift with increasing involvement and leadership from the private sector, often referred to as New Space, taking over a field previously driven by institutional and governmental entities [54]. This resulted in a more dynamic space segment industrial landscape, but it was also more competitive, as cheaper access to space provided opportunities for new entrants. There is also a clear trend toward higher frequencies as a means to address requests for always higher data rates, matching the evolution of the fast-growing terrestrial communication sector. In this section, we provide a review of the space segment, starting with a generic description of current satellite systems, including a particular focus on satellite constellations, followed by a discussion of antenna technologies currently used onboard advanced satellite systems and under development for future satellite systems.

### 2.1. Satellite Description and Classification

The size and mass of satellites have progressed together with the capabilities of launchers. The average “wet mass”, i.e., including propellant, of a satellite has steadily increased from modest beginnings up to about 10 tons in the late 1990s, on par with the capabilities of launchers to geostationary satellite orbit (GSO) [55]. From then on, the development of constellations in Non-Geostationary Satellite Orbit (NGSO), also including Global Navigation Satellite System (GNSS) constellations, and the emerging trend of CubeSats for commercial use, and more generally small satellites, has resulted in a notable reduction of the average mass per satellite. Currently, the majority of satellites launched into space are small satellites [56], referring to satellites with a wet mass typically below 500 kg. This called for a more detailed differentiation between satellite systems, generally following the classifications reported in Table 2, also including examples of commercial satellite systems in respective categories. The list is obviously nonexhaustive, as there are many on-going developments expected to turn into commercial programs in the near future. Some companies, such as GomSpace and Endurosat, provide generic small satellite platforms. The category of femto-satellites is mostly considered these days for educational purposes and laboratory developments, as were CubeSats two decades ago, and may turn in the near future into commercial developments as well. An example of these developments is the SunCube FemtoSat, with a unit size of only 3 × 3 × 3 cm, proposed by the Arizona State University [57]. This is also the case of some PicoSat developments, such as the ThinSat program by Virginia Space, with dimensions corresponding to 1/7U [58]. On the other end of the spectrum, there are a number of satellite developments that are slightly larger than a MiniSat. This includes for example the first generation of O3b satellites (SES) already in orbit and the Telesat Lightspeed constellation under development, both around 700 kg per satellite.

A key parameter in the design of satellites and associated systems is the orbit. This has a significant impact on the antenna design, in particular its directivity and beam steering specifications. Key parameters of typical satellite Earth orbits are listed and compared in Table 3. We distinguished previously between GSO and NGSO. The GSO, also referred to as geostationary Earth orbit (GEO), is particularly convenient for broadcasting applications, as satellites in that orbit have a motion that makes them appear static to a user on the ground. This unique feature is obtained when the orbit of a satellite is in the equatorial plane with an altitude of 35,786 km above the reference geoid. This enables fixed terminals, as often used for example in Direct-to-Home (DTH) satellite broadcasting applications as well as satellite-one-the-pause (SOTP). In the case of satellite-on-the-move (SOTM) applications, the beam steering capabilities are mostly defined by the moving platform (e.g., car, bus) with typically low steering speed requirements. A global coverage is achievable with only three GEO satellites, as implemented for instance with the ViaSat-3 satellite constellation [60]. GEO satellites have however limited performance at high latitudes, where the terminals would be pointing at low elevation angles (typically below 20 degrees). This limitation has triggered the development of Highly Elliptical Orbits (HEO), including the Molniya and the Tundra orbits, characterized with a high eccentricity and inclined orbital planes, providing good visibility over northern regions, such as Russia and Canada. Similar orbits have been considered for southern coverage, specifically Australia. When the satellite is at the apogee, its relative motion to the ground will be significantly reduced, enabling an operation similar to that of a GEO satellite with terminals pointing at a more favorable high elevation angle. Other NGSO include very low, low, and medium Earth orbits (VLEO, LEO, MEO). These are generally circular orbits in inclined planes, although some developments also consider the equatorial plane, such as the first generation of O3b satellites. Inclined orbits are useful to extend the latitude range covered by the satellite. Indicative values for typical altitudes are provided in Table 3. In practice, LEO refers to systems ranging typically from 500 to 1200 km, while MEO generally refers to altitudes ranging from 5000 to 20,000 km. The onboard angular range increases greatly as the altitude reduces, requiring adequate antenna solutions for a proper sizing of the constellation. The visibility time also reduces drastically, indicating that fast steering technology is required for ground terminals connecting to VLEO and LEO satellites, typically imposing electrically steered solutions for both the space and ground segments. Finally, Table 3 also compares typical latency values for the different orbits discussed, considering only the propagation time between the satellite and a user on ground. This is the key parameter that has triggered several LEO constellation developments over recent years, as internet access services and real-time applications are typically not compatible with GEO systems latency, and terrestrial developments on 5G and beyond 5G are placing a particular focus on low-latency solutions.

Other satellite system parameters that have a strong impact on antenna design include the onboard processing capabilities and payload design, which may dictate the number of beams to be produced by the antenna system for example. The adequate sizing of the power management is also critical, as the main parameter in the link budget is the power flux density (PFD), obtained as a combination of the antenna gain and the electronics amplification in transmit. A platform with limited power would require a larger antenna to provide a given PFD, leading to some accommodation issues and associated technological developments (e.g., deployable antennas). Conversely, a platform with higher DC power would require larger solar panels, resulting in accommodation issues, indicating that a good trade-off is needed at the system level. In addition, satellite payloads tend to dissipate a large amount of the available DC power. Thus, platforms with high power available also require adequate thermal control and power dissipation management, including active thermal control in some cases (e.g., active antennas). Finally, another key satellite sub-system having a strong impact on antenna technology is the attitude control. While large satellites generally implement attitude control, with pointing accuracy in the order of ±0.1° for GEO platforms, this may not be sufficient in the case of antenna systems producing highly directive beams. As a rule of thumb, the pointing accuracy is generally specified to be a tenth of the antenna beamwidth to avoid oversizing the performance based on the edge of coverage, including instability. This requires implementing specific tracking systems using beacons on the ground to further improve the pointing accuracy of the antenna, as often used in reflector antenna systems. For smaller satellites, attitude control is not always available. When not present, antennas with quasi-isotropic patterns are generally implemented to ensure a communication link. A solution, also implemented in telemetry and telecommand (TMTC) systems to guarantee a link in case the control of the satellite is lost, consists of using two antennas on opposite faces of the platform with quasi-hemispherical patterns. The following section provides a more detailed discussion of satellite constellations.

### 2.2. Satellite Constellations

A satellite constellation is normally intended as a plurality of similar satellites working together as a system [61]. Unlike a single satellite, a constellation can provide global or near-global coverage, as it can be designed such that from everywhere on Earth (or most of the inhabited surface) at least one satellite is visible at any time. In constellations, satellites are typically placed in sets of complementary orbital planes and connect to a distributed ground stations network on Earth. Depending on the design, they may also use inter-satellite link, in optic or RF [62].

It is possible to classify satellite constellations in different ways, the first is by orbital altitude, for example low Earth orbit (LEO) constellations (e.g., OneWeb, London, UK, Starlink), medium Earth orbit (MEO) constellations (e.g., O3B) or even geostationary orbit (GEO) constellations (e.g., Inmarsat GX, Viasat-3), this usually comprises of a limited number of satellites, typically three or four.

Another way of classifying satellite constellations is by constellation geometry, which is based around satellite positioning and orbit type. This, together with intended service and the limitations of the link budget, determines coverage, which can be global, regional, or targeted. There are a large number of possible useful orbits for satellite constellations, but circular orbits are a popular choice in communication constellations, as all the satellites are at a constant altitude requiring a constant strength signal to communicate and also minimizing the effects of precession [63]. At MEO and LEO, the common geometry types are mainly two: “Walker star” or polar constellation [64] and the “Walker delta” or rosette constellation [65]. A polar orbit is a circular orbit with orbital planes inclined at nearly 90° with respect to Earth’s equator. The orbit is fixed in space, and the Earth rotates underneath. Therefore, a single satellite in a polar orbit provides, in principle, coverage to the entire globe, although there are long periods during which the satellite is out of view from a single observation point on Earth. This limitation, in a polar constellation, is overcome exactly by using multiple satellite equally spaced on the polar orbital planes, providing continuous coverage of the Earth surface by handing over the active communication link from one satellite to the following one on the same orbital plane. In this way, a polar orbit constellation in LEO is naturally providing global coverage of the Earth surface. An example of a polar constellation is provided in Figure 2.

Some LEO and MEO constellations use a rosette design: they are characterized by what are called “inclined orbits” (with inclination substantially smaller than 90°). An inclined orbit constellation provides its best coverage in the areas where the Earth population is concentrated (at latitudes below 45°) but cannot provide a global coverage by itself.

A third way of classifying satellite constellations is by frequency bands used for services, from L- and C- up to Ka- and V-band. The operational frequency band has an impact on the design of the payloads and the link characteristics, and it is usually closely connected with the service that the Satellite intends to provide. Constellations have been extensively used in the past for navigation (e.g., GPS, Galileo, GLONASS), voice telephony (e.g., Iridium), or Earth Observation (e.g., PlanetLabs), which operate typically in the range of the low frequencies, up to L-band and S-band. In most recent years, multiple projects have surfaced aiming at providing broadband internet connection via satellite on a global or near global scale using large scale constellations in LEO and MEO. Despite being theorized a few decades ago, the needed technology to make these massive constellations economically viable has only been developed recently, with the evolution in digital payload and the rise of the new space philosophy causing a revamp in these mega-constellation projects. These constellations operate mainly in Ku- and Ka-bands to maximize the throughput provided and can use even higher frequencies such as V- and Q-bands for their feeder link to the ground station network.

With respect to a GEO communication satellite, an LEO or MEO constellation has some advantages, mainly related to the physical position of the satellites in space, substantially closer to Earth than a geostationary satellite. The reduced distance from the Earth surface is responsible for lower path losses, reducing power requirements and costs of single satellite and Earth user terminals, as well as latency. The reduction in latency enables mission critical communications and high demand applications that are more challenging with GEO and therefore are not yet commonly associated with satellite communications: real-time communications, video chat and videoconferencing, interactive social media, on-line gaming, and some high-end enterprise applications such as remote control (UAVs, terrestrial vehicles, boats), telemedicine, and trading. Another advantage that an LEO constellation has over higher-altitude systems with fewer satellites is that the limited licensed communication frequencies can be reused across the Earth’s surface within each satellite’s coverage footprint. This reuse leads to far higher simultaneous transmission and, therefore, system capacity. The available capacity achievable with the scarce bandwidth available is key in defining the metrics of the constellation and its economic advantage and feasibility, as it plays a major role in lowering the cost per bit of the network.

Compared to a GEO satellite whose orbit is synchronized with the Earth rotation and therefore appears static in the sky for an Earth-bounded observer, LEO and MEO satellites in constellations are constantly moving in the sky. Therefore, the terminal antenna always has to track the satellites in its trajectory across the sky. This means that some of the complexity saved in the space segment is transferred across to the user terminal that has to manage handovers between satellites without dropping the link. For example, depending on the steering approach and complexity of the terminals, the system may implement make-before-break or break-before-make handover. Conversely, LEO and MEO constellation user terminals have the advantage of a better look angle to the satellites, which makes flat panel antennas more suitable for this kind of applications with respect to GEO networks. This topic will be described more in detail in the user terminal section (see Section 4.1).

### 2.3. Satellite Antennas Technologies

Providing an exhaustive list of satellite antenna technologies is obviously impossible in a paper format, and excellent books are already available on this topic [66,67]. The objective of this section is instead to provide a review of key technologies for the applications discussed in this paper and highlight some interesting trends in research. Antennas are generally the most visible sub-systems onboard satellites together with solar panels. Accommodating the antennas to achieve the desired performance while keeping the stowed volume in line with launcher restrictions is often a challenging task, and the type of antennas that may be embarked is often dictated by the platform, or conversely, a mission having specific antenna performance requirements may impose a certain class of platform, either generic or custom-made.

We start this review by addressing first antennas onboard of small satellites. As discussed in Section 2.1, the range of platforms referred to as small satellites is quite broad. small satellites, such as CubeSats and PocketQubes, typically use simple low gain antennas. Commonly encountered solutions include monopoles and dipoles, as well as turnstile antennas operating at relatively low frequencies, e.g., VHF and UHF. These filar antennas are easy to stow in a small volume and can deploy once in orbit using simple mechanisms, providing an antenna size substantially larger than the platform itself. Several products based on filar antennas are available with a generic CubeSat mechanical interface. An exhaustive review of VHF antenna technologies is provided in [68] with particular focus on satellite-based maritime applications. Most of the technologies discussed are applicable to low frequency payloads, in some cases up to L- and S-bands. Interesting solutions under development include a deployable trifilar helix antenna providing a high stowage efficiency [69] and a miniaturized axial mode quadrifilar helix antenna [70]. Fully metallic-folded patch designs with a compact footprint are also reported for microsatellites [71] as well as cross-dipole antennas over an Artificial Magnetic Conductor (AMC) providing a low profile design [72]. These antennas are well suited for communication links requiring low data rates. A breakthrough S-band antenna design providing both beam-steering and polarization agility that advantageously exploits the hosting platform as efficient radiator by resorting to Characteristic Modes Theory (CMT) is presented in [73]. A metasurface superstrate antenna designed with the aid of CMT suitable to be mounted on a single face of a 1U CubeSat platform and operating in the whole Earth Exploration Satellite Services (EESS) frequency band (2025–2290 MHz) adopted for telemetry/payload downlink as well as telecommand uplink is illustrated in [74]. Some solutions are also reported to provide higher directivity from CubeSats typically using higher frequencies, such as X-, Ku-, and even Ka-band. Reflectarrays have attracted some attention as a possible candidate technology, taking advantage of the low stowage volume achievable with flat panels. The first in-flight demonstration of a reflectarray was NASA’s ISARA antenna onboard a 3U CubeSat [75]. This antenna had the particularity of integrating a solar array on the opposite side of the panels to provide enhanced power harvesting capabilities. GomSpace’s GomX-5, a 12U technology demonstration CubeSat developed with the support of ESA and expected to be launched in 2022, will embark a multi-panel X-band reflectarray [76]. Kepler Communications is developing 3U CubeSats embarking Ku-band array antennas, with the transmit antenna having an aperture size of 10 × 20 cm and the receiver antenna occupying an area of 10 × 10 cm [77].

For larger satcom platforms, reflector antennas have been the historically preferred solution. Solid reflector technology provides the best trade-off between cost, performance and reliability. The evolution of GEO satcom payloads, from broadcasting missions in C- and Ku-bands to broadband multiple-spot beam missions in Ka-band, has triggered the development of more advanced feed systems, still relying on reflector-based antenna configurations. The first high throughput satellite embarked a single-feed-per-beam (SFB) antenna system with separate transmit and receive antennas, resulting in a large number of apertures [78]. The development of more compact and integrated feed systems, with dual-band and dual-polarization functionalities, also including a tracking port, reduced the number of apertures from eight down to four or even three [79,80]. Further developments considered more advanced feed arrays with overlapping clusters in a multiple-feed-per-beam (MFB) configuration to further reduce the number of apertures to only two [81,82,83]. This generally comes at the expense of slightly degraded performance due to the sub-optimal cluster excitation. A solution combining polarization-sensitive sub-reflectors and polarizing main reflectors was proposed to obtain the performance of an SFB configuration with only two apertures [84]. An alternative solution considered the use of a dichroic sub-reflector to produce complete multiple beam coverage using a single large aperture [85]. This field of research is still active, as the renewal of existing broadcasting satellites provides an opportunity to embark secondary broadband payloads, and the accommodation of the reflector antennas is always the main limiting factor. The key antenna system parameters highlighting the evolution of broadband satellite solutions are summarized in Table 4. These developments in K/Ka-bands have also benefited from lower frequencies, as multiple-spot beam antennas have been considered at C-band in replacement of more conventional shaped-beam broadcasting antennas [86].

Besides more conventional satellite payloads, there are several developments aiming at introducing higher flexibility through the use of reconfigurable phased array antennas, made possible due to major advances in the field of RFICs. While LEO and MEO solutions, such as Starlink and O3b’s mPower, are mostly direct radiating arrays, GEO solutions still rely on reflector-based imaging configurations to achieve higher gain values. The solutions currently under development, including the OneSat program of Airbus Defense and Space [87] and the INSPIRE program of Thales Alenia Space [88], aim at providing fully reconfigurable software defined payloads based on single-reflector imaging antenna geometries. There is also a trend to use larger reflector apertures to produce higher spectrum reuse over the field of view. With solid reflector technology typically limited to diameters up to about 3.5 m due to fairing constraints, mesh reflectors are being considered as candidate technologies for future missions. Large deployable reflectors typically used in space for missions at lower frequencies, e.g., S- and L-bands, are now being developed for Ku- and Ka-bands, with products available in the 5 m diameter range providing performance compatible with Ka-band operation [89], and much larger diameters are being considered. Besides these developments focusing on the user link, there has also been a number of dedicated activities aiming at providing feeder-link antenna systems at Q/V-bands and above [90]. These can still rely on solid reflector technology but require improved tracking systems adapted to the much narrower beamwidths.

## 3. Airborne Segment

As stated in the introduction, the airborne segment, with its marked properties of flexibility, mobility and versatility, has been considered as an indispensable technology for enabling extremely high data rates and global wireless coverage [45,91,92]. In addition, they represent a more cost-effective solution than satellite layers or the network densification technique applied to the ground level [93]. Furthermore, wireless communication assisted by airborne segments could have many advantages with respect to space segments such as lower transmit power and reduced propagation delay, key features for many applicative scenarios [93].

The idea to exploit flying platforms to reach ubiquitous connectivity is not completely new since the first attempts date back to the 1990s [94,95,96]. However, owing to the recent advances in autonomous vehicles, phased array technology, solar panel efficiency as well as battery UAVs have regained a tremendous attention for both researchers and the industry. For instance, some recent projects focusing on the deployment of UAV platforms for wireless connectivity are reported in [97,98,99].

A straightforward UAV classification belonging to the airborne layer can be performed according to their operating altitude. Specifically, they can be classified into two categories: Low Altitude Platform (LAP) and High Altitude Platform (HAP). However, it is possible to achieve a more detailed classification of these flying platforms according to their size, mission endurance, engine type, take-off and landing methods and wing loading as reported in [100,101].

LAPs can fly at an altitude of tens of meters up to a few kilometers (km), and their greatest strengths are essentially fast movements as well as their extreme flexibility [102,103]. Therefore, they can easily recharge or be replaced if needed. On the contrary, HAPs consist of flying platforms such as gas-filled balloons, airships or aircraft operating in the stratosphere at an altitude of around 20 km [104]. Due to the absence of clouds, thunderstorms and any weather disturbance at these altitudes, solar energy can be effectively utilized and can turn out to be a fundamental asset for HAPs. In general, they are more dedicated to longer missions as well as for providing a wider wireless footprint coverage [105]. Tethered Balloons (TBs) represent another promising technology solution within the framework of LAP flying platforms for delivering wireless communications [106]. TBs consist of inflated balloons, usually filled with helium to lift in air ad hoc equipment, operating in the troposphere at an altitude of around 200–500 m and are tethered to the ground by several ropes [47]. These flying platforms have the potential to deliver a wide variety of wireless communication services by overcoming various challenges faced in terrestrial or satellite segments such as limited coverage area, delay and lack of Line of Sight (LoS) links [107]. Their usage has been exploited in different scenarios spanning from emergency communication systems, observation, military applications and antenna radiation pattern measurements [107,108,109]. Other examples where TBs have also been exploited deal with fronthaul Wi-Fi access by using high gain omnidirectional antennas [110] as well as WiMAX connectivity [107]. Furthermore, in [111], TB has been employed for improving the achievable end-to-end data rate of ground users.

### 3.1. Network Topology

LAPs and HAPs can be deployed in wireless communication networks with different topologies according to the mission needs within which they act mainly as aerial relays or aerial BSs to support wireless communication [45]. In the former case, the flying platforms profitably collaborate with ground BSs or the satellite layer by offering an alternative reliable link by forwarding the incoming data to the recipient. This mode of operation is particularly helpful in emergency situations such as military operations and disaster rescue [49]. Conversely, in the latter case, they play as aerial BS by providing a wide wireless connectivity between ground users and the core network in the absence of terrestrial network or temporary ground station malfunction or maintenance. Moreover, due to their rapid deployment, the airborne segment can help in quickly deploying communication networks after natural disasters such as floods and earthquakes [93,112]. Furthermore, by using HAPs, it is possible to establish a consistent connection between terrestrial users or LAP and satellites constellations, such as CubeSats [73,74] LEO satellites constellations. In addition to the aforementioned UAV applications, data gathering represents another promising use case. By exploiting their versatility and flexibility, they can collect and monitor data from different wireless sensor networks deployed to sense the environment easily and in a cost-effective way.

A possible network partition is represented by non-hybrid or hybrid topology [45]. In the former scheme, illustrated in Figure 3, the flying platforms (i.e., LAPs or HAPS) can work as a BS transceiver or be part of a mesh network of the airborne layer [113], providing a communication link between end users and a core network.

In more details, each UAV, equipped with multiple antenna arrays, is capable of establishing a directional communication link with the different users distributed on the coverage area as well as to provide a wireless communication link with its neighboring flying platforms, hence realizing a flying mesh network capable of improving the overall system performance. This network topology scenario appears to be mainly dedicated to rural zones devoid of terrestrial infrastructures.

In a hybrid topology, as shown in Figure 4, the flying platforms can be fruitfully integrated into an air–ground or satellite–air–ground communication network. They can work both as aerial relays and aerial BS to help the whole wireless infostructure in offering communication services. This communication scheme, crucial for achieving both ubiquitous and seamless connectivity, appears to be the most relevant for future communication systems [43].

In general, it is conceivable to think that the overall wireless communication system could be composed of different smaller wireless networks organized with a dissimilar topology. For this reason, the topology management system represents a challenging task to tackle in order to reach superior systems performance as well as to guarantee the desired QoS in future applicative scenarios. In this framework, AI and ML technology will represent a fundamental resource within the network management and automation as well as meet the reconfigurability demand [29].

### 3.2. Spectral Efficiency Improvement

In future wireless communication generations, airborne communications are expected to play a prominent role in the delivery of next-generation services. The UAVs acting as flying platforms can provide a reliable aerial access link to different ground or satellite users in different scenarios such as temporary ground stations disruption, hotspot areas or large public venues—scenarios in which many users strain the available wireless resources [114]. Therefore, efficient wireless communication technologies are essentially for serving multiple users and ensure the desired QoS. The Multiple-Input–Multiple-Output (MIMO) technique represents a possible wireless technology strategy that can improve network performance by exploiting both the Diversity Gain (DG) and the Multiplexing Gain (MG) [4]. A MIMO system with the corresponding MIMO channel matrix *H*, offers *K = rank (HH*)* parallel Single-Input–Single-Output (SISO) subchannels with different gain where it is possible to send different streams of data. In general, two scenarios are possible: absence of knowledge of the Channel State Information (CSI) at the transmitter and knowledge of the CSI at the transmitter (CSIT). In the former case, the transmitted power is evenly split into each *K*th subchannel. Conversely, in case of CSIT, it is possible to obtain an optimum power allocation for each *K*th subchannel (water filling technique) according to the eigenvalues (*λ_i_*) of the MIMO channel matrix *H* in order to optimize the spectral efficiency. However, in practice, perfect CSIT is not possible due to errors on channel estimation and feedback delay [115]. MIMO performance in case of feedback delay and channel estimation errors have been analyzed in [116,117,118]. For the airborne layer, the CSIT concern turns out to be even more pronounced due to the high altitudes and the movements of LAPs and HAPs flying platforms [47]. An interesting solution for the CSIT mitigation has been proposed in [119] where a TB relay and an effective interference alignment scheme for maximizing the spectral efficiency of HAPs-ground stations communications were presented. Moreover, a virtual MIMO (V-MIMO) system, realized by connecting multiple HAPs, has been proposed in [120].

Another attractive technology that could be exploited by aerial platforms for improving both Spectral Efficiency (SE) and Energy Efficiency (EE) is represented by massive MIMO technology [6,7,8,9] capable of serving multiple users simultaneously in the same time-frequency resource through smart array antennas with multibeam radiation pattern [6,7,8,9]. Specifically, in [121,122], the authors reported some examples of massive MIMO applied to HAPs, whereas the potential of massive MIMO systems for communication with UA- based LAPs are illustrated in [123,124,125].

In addition of the aforementioned wireless communication techniques, Full-Duplex (FD) technology represents a promising solution to meet the tremendous increasing system requirements as well as a viable alternative in addressing spectrum scarcity [126]. More in detail, an FD wireless terminal is capable of transmitting and receiving simultaneously in the same frequency band by allowing, theoretically, to double the SE with respect to conventional Half-Duplex (HD) systems [127]. However, one of the biggest impediments of FD communication that leads to undermining hypothetical SE doubling is the presence of wireless interference. Due to the simultaneous uplink and downlink wireless communications, it is possible to generate interference to adjacent users or BSs and, at the same time, receive interference from them [128,129]. Figure 5 shows an example of both HD and FD wireless communication. Specifically, an HD system characterized by a separate resource (frequency band or time), highlighted by different arrow colors, between the backhaul link (black arrows) and the access link (green arrows) is shown in Figure 5a. Conversely, Figure 5b emphasizes an FD scenario where both the backhaul link and the access link share the same frequency band or time resource.

In the case of an FD scenario (Figure 5b), it can be seen that the two interference topologies due to the collaboration among different devices should be accurately addressed in order to reduce the performance degradation of the overall system, precisely, the interference within the same transceiver, also known as Self-Interference (SI) as well as the interference coming from neighboring users, identified as access or backhaul interference. FD communications can be successfully implemented if each FD device is capable of guaranteeing a sufficient SI cancellation (SIC), namely a satisfactory transmitted signal attenuation below a certain threshold in order that it does not create a problem for its receiver. An extensive overview about hardware and software SIC is reported in [128]. Concerning the interference coming from the simultaneous communications of other users, it can be accurately reduced by minimizing the radiation pattern lateral lobes in the direction of other users through advanced beamforming techniques [12,13,14]. Although FD wireless communication has attracted much attention in UAV-assisted wireless communication [130,131,132,133], recently, Hybrid-Duplex (HBD) communication has triggered enormous interest [134,135,136,137,138]. It consists of a wireless network where both FD and HD devices are involved, as depicted in Figure 6. More in detail, in Figure 6, FD technology is implemented only at the ground segment base stations (FD-GS), whereas the airborne segment operates in HD mode (HD-AS). A separate resource (time or frequency) is dedicated for the uplink and downlink signal related to the UAVs (highlighted by different arrows color), whereas for the ground segment BS, they share the same resource. This choice seems to be plausible since the SIC turns out to be easier to tackle at ground level rather than at an airborne one as well as from the energy point of view.

### 3.3. Airborne Antennas Technologies

In general, a flying platform is equipped by many electronic components that can be grouped into three main subsystems [27]: flight control, energy management and transceivers. The flight control subsystem, composed of some sensors and actuators, is responsible of platform stabilization and mobility. The energy management subsystem handles the energy and its storage by using solar panels and batteries, overall being responsible for available energy. The transceiver subsystem represents the set of electronic components that allow one to transmit and receive data. According to the mission and the application purposes, different equipment and technologies can be adopted into these onboard subsystems. In this subsection, one of the most important components of the transceiver subsystem is discussed, namely the radiating system. Antennas are certainly among the fundamental components of UAVs, and they are determinant for the performance of the onboard transceiver subsystem. Therefore, high gain, high efficiency and low-profile airborne antennas represent some key requirements to consider during the design phase. For example, an antenna array composed of four printed monopole antennas working at 2.4 GHz embedded in the structural components of a UAV wing is proposed in [139]. An efficient radiator composed of compact and low-profile probes accurately placed on the UAV body has been designed in [140] by exploiting the Characteristic Modes Theory (CMT) [141,142,143]. In [144], a broadband slotted blade dipole antenna is described. A conformal phased array antenna for UAVs with wide scanning range is presented in [145]. Some solutions regarding the design of radiating systems for HAPs are illustrated in [146,147,148].

Concerning the coverage area, the flying platforms scenario differs from that of ground segment, whose coverage is typically rectangular in the *uv* plane, since a circular scan area turns out to be more appropriate [91]. Then, within a predefined circular scenario, there could be several possible cells configurations. Figure 7 illustrates two examples of cells configuration within a circular scanning area with a maximum coverage angle of sin(*θ_max_*) off broadside direction (*θ* = 0°), namely with *n* = 8 cells (Figure 7a) and *n* = 20 cells (Figure 7b).

The cell configuration scheme has a strong impact on the UAV antenna parameters such as the Half Power Beam Width (HPBW) on both horizontal and vertical planes as well as lateral lobes.

As previously stated, 5G, 6G and future wireless generations open the door to mmWave communications. However, owing to a deeper propagation loss and higher sensitivity to obstacles, they have to cope with a coverage limitation when compared to sub-6 GHz communication systems. Therefore, active electronically beam-scanning antenna arrays represent a pivotal technology for the air segment to provide high gain capable of counteracting high path loss, offering low interference communications as well as concurrent multibeam radiation patterns. However, in the case of mmWave, the radiating system design turns out to be even more important due to the significant losses of phase shifters and a lower Power Amplifiers (PAs) efficiency [149] that lead to a more complicated thermal management [150]. From the energy point of view, passive cooling systems are preferred to active ones by the industry since they do not need electricity. In the framework of antenna array design, the simplest way to help the cooling system to dissipate heat is to increase the distance among antenna elements [20]. However, increasing the inter-element spacing too much could lead to grating lobes or high lateral lobes inside the visible region, with a harmful interference effect in a multiusers scenario. The most popular array layouts are organized in square or rectangular lattices. However, the benefit of adopting a triangular lattice in a massive MIMO scenario by providing a superior angular resolution as a function of the antenna beam steering is presented in [15,151,152]. An alternative approach using a triangular lattice of beams has also demonstrated interesting performance in array design with beam-switching operation [153]. A Ka-band phased array for HAPs application composed of open-ended substrate-integrated square waveguides and a 4-channel beamformer circuit produced by Anokiwave was described in [154]. A relevant mmWave beam steering 8 × 8 array design solution operating from 26.5 to 31 GHz for 5G BSs based on gap waveguide technology is presented in [155]. Low loss feeding, high gain and exceptional thermal handling are guaranteed by an all-metal multi-layer assembly. Advances in 3D-printed technology and manufacturing processes make Dielectric Resonator Antenna (DRA) technology another attractive solution for the development of commercial array antennas at mmWave [156]. For instance, reference [157] presents an 8 × 8 array based on DRA fed by a slot antenna operating within a 5G wireless communications mmWave frequency band. An extensive overview of available antenna array technologies for mm-Wave communications is reported in [158].

With the purpose to reduce both cost and power consumption, key factors for future wireless communications, unconventional arrays designing such as sparse arrays [159,160], thinned arrays [161] and subarrays techniques [162,163] will represent a noteworthy airborne-array-designing technique in the future. However, achieving the same Equivalent Isotropic Radiated Power (EIRP) of a classic array—namely, each radiating element arranged on a regular and periodic lattice equipped with a Transmit/Receive Module (TRM) able to control both amplitude and phase of the signal—requires that the unconventional arrays’ Power Amplifiers (PAs) have to provide a higher output power. This aspect introduces new challenges at the system level due to a greater tendency of PA nonlinearities that can affect the Error Vector Module (EVM) or the Adjacent Channel Power Ratio (ACPR), namely the modulation error of the signal with respect to the reference constellation and the users interference operating in the adjacent channels [164]. To overcome this issue, some linearization techniques such as the Digital Predistortion (DPD) [165] can be successfully adopted in order to maintain transceiver linearity as compliant with the systems’ requirements.

Another crucial aspect that phased array designers must face is calibration [166,167], which allows one to balance some manufacturing errors and electronic inaccuracies (e.g., TRM amplitude and phase unbalance) capable of approaching the array theoretical radiative performance such as gain and side lobe level reduction. Some alterations of both the amplitude and phase of array elements feeding inevitably degrade the beamforming quality and hence the link data rate. However, array calibration represents one of the main array costs, and hence, its usage must be accurately assessed by making a sort of tradeoff between the desired performance and overall cost [168]. For instance, within the framework of 5G, many phased arrays without the calibration procedure have been proposed [168,169,170,171] with the purpose to drastically reduce their cost by highlighting acceptable array performance degradation. Some phased array calibrations methods are described in [172,173].

Despite the advantages of mmWave communications, such as larger spectrum, the adoption of large phased arrays for both UAVs and user mobility makes the antenna beam alignment between transmitter (TX) and receiver (RX) a challenging task to be tackled to guarantee the link robustness and hence satisfy the expected QoS [113]. It is necessary to determine the best TX and RX beam pair for reliable communication. A beam alignment solution is represented by resorting to training and tracking scheme [174] by identifying the best beamforming array feeding among all beam direction combinations. However, if highly directive beams are adopted both at the TX and RX side, the wireless communication system will suffer a large beam setup time. To overcome this issue, the adaptive beamwidth approach has been proposed [175]. First, the TX and RX find their angular sectoral by using a wide beam. Then, the beam alignment management narrows down their beamwidth gradually to reach their maximum directivity. Other solutions are based on a combination of both mechanical adjustment for coarse alignment along with a fine beam tuning with electrical adjustment as proposed in [176].

In the mmWave and sub-terahertz domains, quasi-optical antenna solutions are also considered as a promising alternative to reduce the number of control nodes while keeping high gain figures [48]. In this respect, geodesic lenses have attracted some attention for their highly efficient fully-metallic design implementation [177]. Metamaterials are also considered to be a promising avenue to further enhance the performance of array designs, addressing their miniaturization and inter-element coupling mitigation [178].

## 4. Ground Segment

Satellite communication has the potential to gain a big share of communication market, as it enables services that are not achievable via cable, such as mobility or connection from remote or rural sites. As the demand for these services grows, the demand for broadband satellite communications is also growing, and this is one of the reasons why many new high-capacity satellites and constellations are now in the making.

Considerable focus in Satcom technology is given to what happens in space, but what happens on earth is as important. Every satellite, no matter how advanced, is still only a part of a larger system and a satellite or constellation, and to be correctly exploited, it needs an adequate network of gateway ground stations and user terminals. In particular, the user terminal is key in the success of the satcom network, as it will impact the penetration into the market and will make the network successful and sustainable from an economic point of view.

Many of these newer satellite systems we are seeing in development are NGSO constellations, made of smaller satellites but comprising hundreds or thousands of them, adding significant complexity to the communications system. While a GEO orbit is synchronized with the Earth rotation, and therefore the satellite appears static in the sky for an Earth-bounded observer, NGSO satellites arranged in constellations are constantly moving in the sky, adding tracking, Doppler shift and handover complexity to both the space and the ground segment. Moreover, NGSO constellations need to rely on extremely big networks of ground stations, as every satellite in the sky needs to be in view of a gateway ground station. Intersatellite links (either optical or RF) can ease the pressure on the ground network by removing the need for a satellite to be constantly connected to a ground station, but it is also adding complexity to the routing of the data and adding constraints and costs to the design of the satellite.

This added complexity in NGSO satellites systems however comes with some advantages with respect to a GEO satellite, advantages that impact the design of the satellite itself but also, massively, the usability and effectiveness of user terminals. These main advantages are:(a)The lower altitude in the sky means that the required performance to establish the link is lower, as the free space loss is drastically reduced. This means smaller satellites, less power and smaller antennas both on the ground and in orbit. A smaller antenna for a User Terminal represents a major advantage.(b)The lower altitude also reduces latency, thus making satcom networks comparable with ground networks (especially for LEO systems).(c)The fact that the satellites are constantly moving in the sky means that the look angle from Earth to the satellite is constantly changing, and most of the time, it is in an advantageous position, approximately overhead of a user. In a geosynchronous system, moving toward northern latitudes in the northern hemisphere (and the same southern for the southern hemisphere) means that the look angle reaches lower elevation values, making the link budget harder and harder to close.(d)Moving satellites in the sky means that the impact of blockage from buildings, mountains, terrains, etc., is massively reduced as the look angle constantly changes, naturally avoiding obstacles.

Taking advantage of these assets of low orbit systems is key in the success of the constellation model and is where the satellite industry must invest to transform NGSO communication in a sustainable reality alongside the more mature and proven GEO Systems.

### 4.1. User Terminal Antennas

Historically, a GEO system User Terminal (UT) is made of a parabolic antenna plus an antenna control unit mounted on a fixed structure on top of a building [179]. As the antenna is looking at a fixed point in the sky and may require achieving low elevations with respect to the zenith, a parabolic antenna is well suited for the task, guaranteeing a steering capability for pointing, a good performance at any steering angle (the well-known key-hole limitation at the zenith can be affectively mitigated for GEO terminals) and a relatively low cost and high reliability.

When mobility came along however, a traditional parabolic antenna was not the best option for all markets anymore, as it is bulky, heavy, fits into an unappealing dome and not fulfilling the requirements of a terminal that needs to be mounted on a possibly small moving, or flying, vehicle. The need for compact low-profile antennas for mobile terminals contributes to the development of more compact (and complex) geometries for steerable reflector antennas [180], which have the capability to fit into a smaller, more compact volume and to maintain contact with a GEO satellite while the vehicle is moving. These antenna designs are usually extremely expensive, given their complexity, and therefore have a quite limited market, mainly limited to high-end satellite communications (trains, big vessels, commercial airplanes, etc.).

When referring to NGSO UTs, antennas need to continuously track moving satellites in the sky. The continuous tracking adds a significant mechanical stress to the reflector antenna motors with respect to a traditional GSO user terminal. LEO tracking antennas must also move rapidly, as a typical LEO satellite can stay in the visibility span of a user terminal (up to 120 degrees typically) for as little as 10 s. This makes traditional reflector antennas not particularly suited for LEO applications.

A shift in the paradigm of the UT came along with the introduction of flat panel User Terminals, integrating flat panel antennas in their enclosure. Flat panel antennas have the potential to be more integrated into mobility platforms, but this is not all: being smaller, flat, fast tracking, less expensive and immune to mechanical stress, opens the door to markets that have not been touched by satellite communication before. These characteristics are rather important for a GSO system UT, but are utterly fundamental for an NGSO system, making the flat panel UT the Holy Grail for the success of low orbit satellite systems.

The challenges in developing a flat panel antenna for Satcom applications are numerous and span from the engineering aspect to the marketing and regulatory [181]. The most challenging design goals for the flat-panel antenna are the trade-off between performance, power consumption, bandwidth, aperture efficiency, reliability, and manufacturability. Performance at low elevations (due to steering loss resulting from the projected aperture) is also a major limiting factor, especially for GSO systems (NGSO UTs have satellites approximately overhead for most of the operational time).

Most flat panel systems also suffer regulatory issues, as many existing regulatory requirements for Satcom user terminals are historically based on the parabolic-type antenna technology with invariant radiation patterns over antenna steering and more stringent side lobe level requirements. For the flat-panel type antenna, however, the radiation patterns are changing with beam steering, and it requires substantially more design efforts for a flat panel antenna to comply with a typical radiation mask.

The first solutions for flat antennas to arrive on the UT market were mostly hybrid solutions at low frequencies (L-band, X-band) for the GEO market, combining electronic steering with mechanical pointing. With the advent of NGSO and higher frequencies, the challenges have increased due to required miniaturization, increased operational bandwidth and the need for faster 2D tracking. In recent times, the most popular flat-panel antenna solutions for broadband satellite communication are phased arrays using either analog, digital or hybrid beamforming techniques. These antennas are also commercially known as Electronically Scanning Antennas (ESA).

Analog beamforming is a relatively affordable (USD ~1.5 per element) and low-power solution, but the antenna performance usually struggles with broad bands due to the frequency dependence of the integrated phase-shifters, generating distortions in beamforming away from the designed center frequency. Conversely, digital beamforming is more flexible and can be performed over wide bandwidths due to its intrinsic use of true time delay, which ensures frequency-independent behavior [5]. The digital beamforming processor can be extremely power hungry and challenging from both cost and performance point of views, especially for high frequency bands such as Ka. Hybrid Beamforming combines aspects of analog and digital beamforming, achieving a lower power consumption but still maintaining some of the flexibility given by the digitalization.

Passive beamforming solutions are also being developed with the aim of achieving a better trade-off between performance and power consumption, which is considered of high importance in some Satcom market applications. The passive beamforming arrays usually have significantly lower DC power consumption than active arrays. In the range of passive beamforming, metamaterials and metasurfaces are currently used to design flat panel antennas. Metamaterials are artificial structures with electromagnetic properties that cannot be obtained in nature and can be used in an antenna to steer the beam without the use of complex Beam Forming Networks (BFNs), by locally tuning the reflective index with discrete low cost/low power active elements such as diodes [182,183]. The use of metamaterials however poses some challenges, mainly linked to the resonant nature of the design: they normally exhibit low bandwidth, high losses and a relatively small steering range.

Liquid-crystal (LC)-based passive beamformers have been designed for Ku-/Ka-band UTs [184]. This design is based on the principle of phase delay in a planar transmission line. It is possible to introduce a phase delay to a signal on the transmission line by controlling, with the application of a DC voltage bias, the alignment of the LC molecules in a LC substrate, causing a change in the local dielectric constant. While this design presents improvements in operational bandwidth with respect to a traditional metamaterial-based design, it is subjected to the intrinsic slow response of LCs and may result in slow beam steering and switching, especially at low temperatures, as LC response time is temperature dependent.

Another solution successfully used on the market is represented by Variably Inclined Continuous Transverse Stub (VICTS) antennas [185], which is a hybrid mechanical/electronic design combining stacked radiating surfaces with rotating motors. Different RF design of the disks and different rotation methods can be used [186,187], achieving different degrees of compactness and RF performance, but generically VICTS antenna design are characterized by a wide scan angle, reduced steering losses and low power consumption. This technology is usually less low profile and heavier than an ESA and is subjected to the drawbacks of integrated moving parts (motor reliability, usage, etc.). An example of other hybrid mechanical–electronic designs are presented in [188].

Lastly, microwave lenses can also be used to design steerable antennas. An example of antenna design including lens is presented in [189]. The base design is still an active phased array, but the lens work as an optical beamformer reducing the complexity of electronic BFNs, and therefore actively reducing the overall number (and therefore cost) of electronic components and the power consumption of the antenna. Furthermore, the optical properties of the lenses can be used to reduce the scan loss, achieving better performance at low elevations. Conversely, lens antennas are usually challenging from a form factor point of view, in terms of low profile and weight, and cost, limiting the usage in some markets. Fully passive solutions using printed circuit board (PCB) technology to produce phase-shifting surfaces have also been described with a centralized feeding point [190,191] or a printed radial slot array [188] in an attempt to produce low-cost solutions at the expense of a reduced integration.

Different satcom markets are normally characterized by distinct requirements and priorities and therefore numerous design approaches and technologies can be successful at the same time, as they may address different needs.

### 4.2. Gateway Antennas

All satellites require gateways to connect to the core network and exchange data between users. Gateway stations (or ground stations) provide the interface between the satellites out in space, and the terrestrial networks for public switched telephone networks, cellular networks and data transmission networks.

A gateway station consists of several different components that allow transmission and reception to and from the satellite, amplification of the signals, transformation and connection to the terrestrial network. The main part of a ground station is the antenna that sends and receives the satellite signals. Ground station antennas are typically parabolic dishes pointing to one single satellite each. Depending on the frequency, gateway antennas vary in size and complexity. For lower frequencies, they are generally in the order of 10 m diameter and decrease in size for higher frequencies. Generally speaking, the higher the frequency is, the smaller the antenna is, and the harder it is to point the antenna to the satellite. With GEO satellites, the task of pointing and maintaining the link to the satellite is simplified by the fact that the satellite is static in the sky; thus, the gateway does not need to track the satellite movements across the sky. NGSO gateways are more complex systems from a ground network perspective, as they need tracking antennas, handover between subsequent satellites on the orbital arc, and tracking of multiple satellites from the same site. Therefore, while traditional parabolic dishes are generally effective for gateways communicating with GEO satellites, they are limited when it comes to tracking fast-moving LEO satellites.

The main problems associated with the traditional parabolic dish approach for a NGSO gateway station are the motorization of the antennas and the large footprint of the gateway station. The large footprint is due to the large number of separate reflectors needed (one LEO gateway station can track up to 15 satellites at the same time) and the need to guarantee enough distance between antennas to avoid line-of-site (LOS) issues between them, which is likely to happen during tracking, especially for low elevation angles over the horizon (as an example, see OneWeb Satellite Network Portal [SNP] in Figure 8). The antenna mechanical steering and the issues associated with it (need for frequent maintenance, reliability) is another limiting factor of traditional gateway stations.

In this perspective, technologies are emerging aiming at applying the principle of flat multibeam antennas to gateway antennas as well. An electronic steerable gateway is one way to mitigate the previously mentioned issues. Motorization systems can be avoided completely, and the footprint can be reduced by using a single structure to track all the satellites in view. This requires an electronically steerable antenna technology that can dynamically establish a high number of simultaneous beams with a reduced ground infrastructure. The reduced footprint could also allow the installation of the gateway near to the backhaul control center nodes instead of far remote areas where land is available at a low price, thus further reducing the terrestrial network latency.

Various architectures have been proposed for multibeam gateway antenna systems. To be able to achieve multi-beam behavior on a wide scan angle needed to track as many satellites as possible up to low elevation angle over the horizon, the gateway system is usually made up of a multifaceted structure combining different flat antenna panels distributed at different angles with respect to the ground, in the shape of a dome or similar. Some design solutions and recent commercial offerings are proposed in [192,193,194,195,196].

## 5. Application Scenarios

In this section, relevant aerospace scenarios described in the literature will be reviewed, along with network architectures supporting those scenarios. As anticipated, the SAGIN paradigm, sometimes referred to as Space Information Network (SIN), should be considered as the main reference [197,198], encompassing the challenging interworking of space systems, aerial networks, and terrestrial communications. Resources in the three network segments are limited and unbalanced [197], thus requiring careful design for the integration to be successful. The investigated scenarios and enabling components are summarized in Table 5 and are discussed below.

Some common application scenarios by leveraging SAGINs and key services offered by the network infrastructure at different layers of the SAGIN are graphically illustrated in Figure 9. Focusing on the current 5G deployment [210] and the ongoing work for the definition of the 6G standard [211], it is evident that different radio access technologies, also including the satellite component, are needed for such integration. Terrestrial services can be augmented with the development of VHTS systems and LEO mega-constellations to meet stringent requirements, such as high bandwidth, low latency, and increased coverage.

The work in [210] focuses on the role of satellites in 5G networks, highlighting that enhanced mobile broadband (eMBB), for which user data rates and spectrum efficiency are crucial, and massive Machine Type Communications (mMTC) are to be considered as common scenarios, in which the satellite plays a role as backhaul to interconnect separate parts of the same 5G network. In the case of mMTC, the ability to support a multitude of connections is fundamental, distributed over time and frequency, each exchanging few data packets. Additionally, satellite systems may strongly support delay-tolerant services requiring high reliability and high availability [210]. The work in [212] highlights that the terrestrial infrastructure, in its current state, may be insufficient to guarantee 5G Key Performance Indicators (KPIs) in some scenarios, for instance, in providing ubiquitous coverage, or in the case of infrastructure unavailability, thus requiring the use of aerospace solutions to increase both the resilience and the availability of the network, in turn improving the Quality of Experience (QoE) perceived by users. This is particularly true for IoT scenarios [198,212,213] in which both resilience and network availability may be key requirements, as in the case of smart grids [200]. The different roles and equipment envisaged for IoT devices in 5G scenarios are analyzed in [212] when considering the joint use of satellites, UAVs, and ground nodes, proposing UAVs to act as 5G User Equipment (UE), as base stations (5G-gNBs), or as transparent relay nodes. Satellites, especially LEO ones, can act as 5G-gNBs or as relays depending on the payload (regenerative or transparent, respectively). The case of future 6G networks is considered in [211], emphasizing that the SAGIN network paradigm will become even more central in upcoming developments, and underlining how the combination of Artificial Intelligence (AI) and Software Defined Networking (SDN)/Network Functions Virtualization (NFV) will enable zero-touch orchestration, optimization, and management of networks.

The upcoming 6G standard, still in its definition phase, expands the service classes foreseen in 5G. According to [19], the ones to be added are the so-called Mobile Broadband Reliable Low Latency Communication (MBRLLC), the massive Ultra-Reliable Low Latency Communications (mURLLC), and what the authors dub as Human-Centric Services (HCS), and multi-purpose control, localization, sensing, and energy services. The latter two are classes comprising a vast group of applications, such as multisensory extend reality or even wireless Brain-Computer Interactions (BCI). Some services and some scenarios will require on-demand capacity to be deployed, or on-demand coverage in poorly covered or too busy areas. Because of this, the use of LAPs or HAPs according to the area and to the requirements is seen as a necessity to support the ground infrastructure when and where needed. Due to the integration of ground and airborne networks, communications must be supported in 3D space, accounting for the additional degrees of freedom because of the different heights of LAPs and HAPs, if not even satellites. Such a complex interplay has the potential to support existing services and open to new ones. For instance, the paradigm of autonomous driving is attracting increasing attention all around the world, and for it to be a reality in every corner of the world, satellite access is likely crucial, providing access to the network in poorly covered areas (see the case of rural ones), real-time maps updates and additional services, such as safety-related ones [18,26,50]. Furthermore, the idea of smart cities strongly relies on 3D communications, with much potential for UAVs to provide coverage extension services, on-demand bandwidth, monitoring services, and mobile crowdsensing [26], among others. The white paper in [209], discussing of an EU vision of the upcoming 6G network ecosystem, describes NTN nodes as “computing and storage in the sky” for task scheduling, task offloading, and caching capabilities [26,50,209,214]. Generally speaking, NTN nodes can be seen as data centers in the air, which are supposed to strongly leverage AI-based techniques [206,207].

SAGINs can also be described through the lens of service-oriented networks [50], in which the focus is moved from coverage, user access, and data exchanges to the possibility of offering guaranteed services to final users. This means that on-demand reconfigurability must be possible to tailor the network configuration at any time to adapt to the requirements of the services to be provided. SDN and NFV are key technologies in this matter, and flexible components, such as UAVs, are crucial to recompose the so-called *service function chain* accordingly to the considered requirements. A 3D network architecture with moving elements poses several challenges in terms of mobility management: node movements must be carefully considered and predicted to minimize, e.g., link interruptions that impact user services.

Another research line that has seen a recent revamped interest in the scientific community is represented by indoor localization, more precisely by the possibility to provide services offering a continuum indoor-outdoor localization and positioning [204,205,215]. SAGINs have the potential to support both localization and positioning and location-aware services, which will be of paramount importance for autonomous vehicles and in environments in which purely GNSS-based services cannot work (e.g., indoor, urban canyons) [205]. The case of autonomous vehicles is challenging from several viewpoints, especially looking at deployment, coverage, and capacity issues of the roadside infrastructure [202]. The network comprising both autonomous vehicles and roadside units is referred to as Internet of Vehicles (IoV), in which services similar to real-time autonomous driving assistance, collision avoidance, and traffic management, among others, are key services to be made available [203]. Those services require real-time data exchanges in most of the cases, thus calling for the use of edge-cloud computing in a synergic manner [50], offering low delay, caching, and offloading capabilities at the edge, complemented by significant storage capabilities and computational power at the cloud level.

The paradigm of IoV is inspired by IoT, which sees a plethora of application scenarios of interest described in the literature, especially when considering the interplay of UAVs and satellites. IoT is described as the means to collect data from sensors or RFID [216,217] and to send control messages to actuators in [200]. The assumption is that the smart objects are remote, dispersed over a wide geographical area, or are inaccessible; thus, the airborne segment is a viable if not the only option to connect them. The concept of IoT is specialized into what the authors define as Internet of Remote Things (IoRT) [200], and it is of interest for smart grids, environmental monitoring, and emergency scenarios. Several additional scenarios can be read in [212], such as military ones for dull, dirty, and dangerous operations; or in the case of disasters for recovery and support operations, as performed in Haiti in 2013 for goods delivery and for providing temporary connectivity because of the unavailability of the terrestrial infrastructure; for real-time traffic monitoring, as also proposed in [218], to assist in the case of heavy road congestion; finally, to enable local weather forecasting and monitoring [219], removing the need for fixed stations. The success of UAVs can be explained by the increasingly low prices, among other factors, which makes them an ideal option for several applications, such as fire detection and control or search and rescue operations [105], in addition to those already mentioned above. Other IoT scenarios of interest are covered in [220,221], such as power line inspection, monitoring of cultural heritage sites, and smart farming [214,222], all involving the use of UAVs. An interesting perspective is provided in [206], which foresees the use of UAVs to provide near-user edge computing capabilities in IoT scenarios in which edge and cloud infrastructure may be unavailable, and satellites for cloud computing capabilities. Complementary solutions to UAVs, which fall into the category of LAPs, are described in the literature in the form of HAPs, such as balloons [198]; although less used in real deployments, they offer wider coverage and longer endurance. Because of those features, HAPs are preferred when it comes to providing reliable wireless coverage in large geographic areas [105]. Network architectures for SAGINs, thus involving LAPs, HAPs, and satellites at different orbits, are described in [197,223], emphasizing the achievable level of QoS. Three reference scenarios, i.e., search and rescue, surveillance and monitoring, and goods delivery, are mentioned in [223] involving a Flying Ad Hoc Network (FANET) and nanoSATs. The case of Non-Radio-Line-of-Sight (NRLoS) conditions in a dispersed FANET is covered in [213], foreseeing a back-haul via satellite to deliver data.

## 6. Conclusions

A comprehensive survey regarding recent advances and technical solutions in the design and development of breakthrough space–air–ground-integrated networks for supporting seamless and ubiquitous wireless connectivity for future 6G wireless communications has been carried out. The paper opens with an extensive overview about the space segment by focusing on satellites classification, constellations, and current and future trends on antenna technologies. Then, a detailed investigation regarding the air layer is provided, and its prominent role in the delivery of next-generation services is described and discussed. Moreover, particular attention is also paid to the ground segment focusing on both user terminal and gateway antennas. Finally, relevant application scenarios regarding the paradigm of SAGIN in present and future wireless communications are discussed, covering 5G, B5G and 6G use cases.

## Figures and Tables

**Figure 1 sensors-22-03136-f001:**
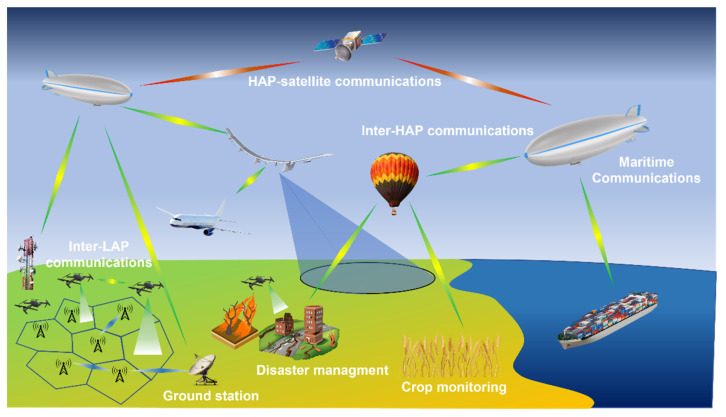
Example of a VHetNet scenario by considering some space, air and ground network components as envisioned in 6G wireless communications.

**Figure 2 sensors-22-03136-f002:**
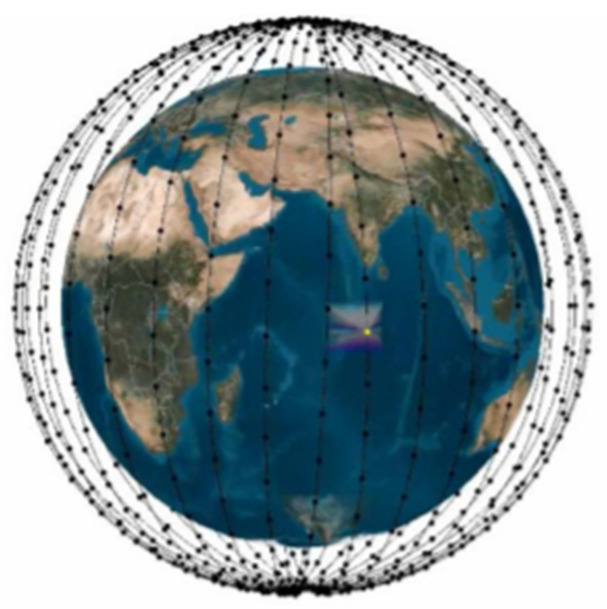
Constellation pattern of OneWeb system: 648 satellites distributed across 12 circular orbital planes at an altitude of 1200 Km; each plane inclined at 87°.

**Figure 3 sensors-22-03136-f003:**
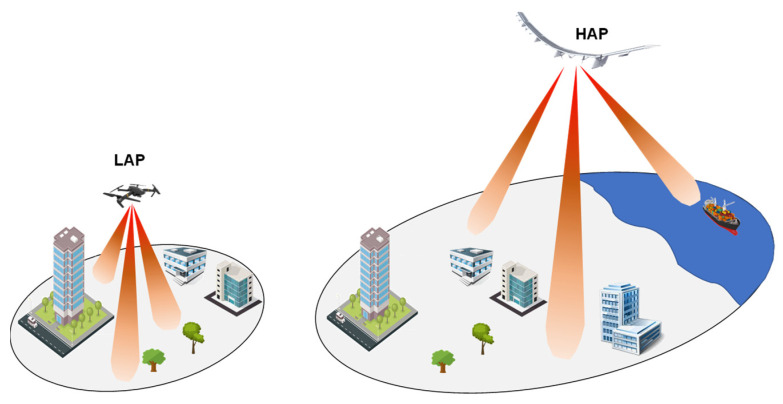
Example of non-hybrid network topology by involving the airborne segment only.

**Figure 4 sensors-22-03136-f004:**
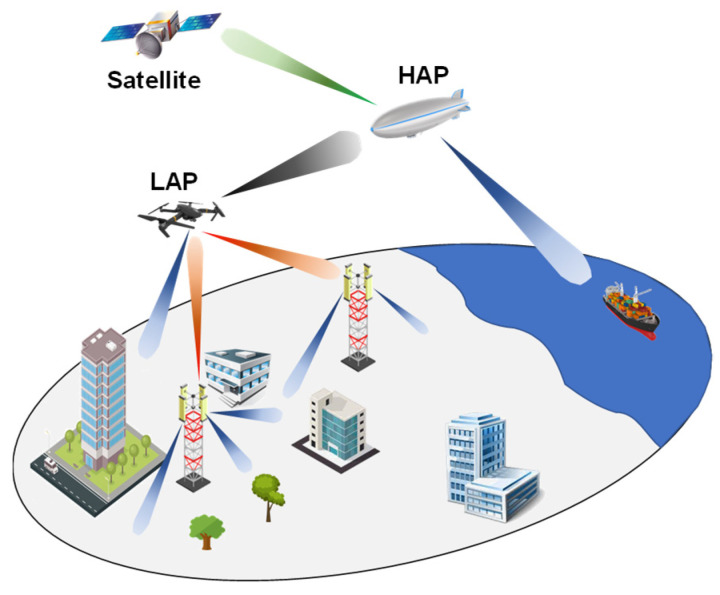
Example of hybrid network topology by involving terrestrial, airborne and satellite segment.

**Figure 5 sensors-22-03136-f005:**
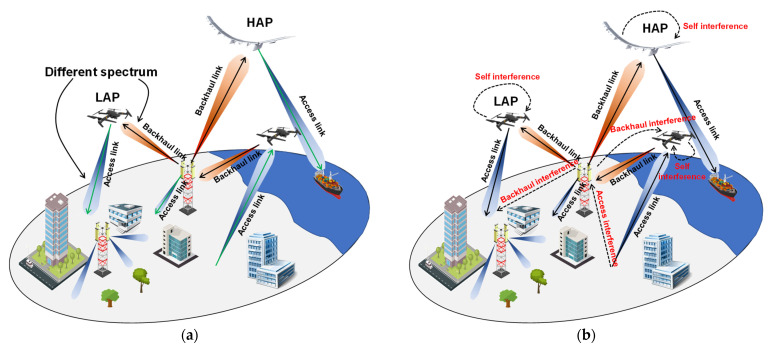
Example of (**a**) Half-Duplex (HD) and (**b**) Full-Duplex (FD) wireless communication through ground and airborne segment.

**Figure 6 sensors-22-03136-f006:**
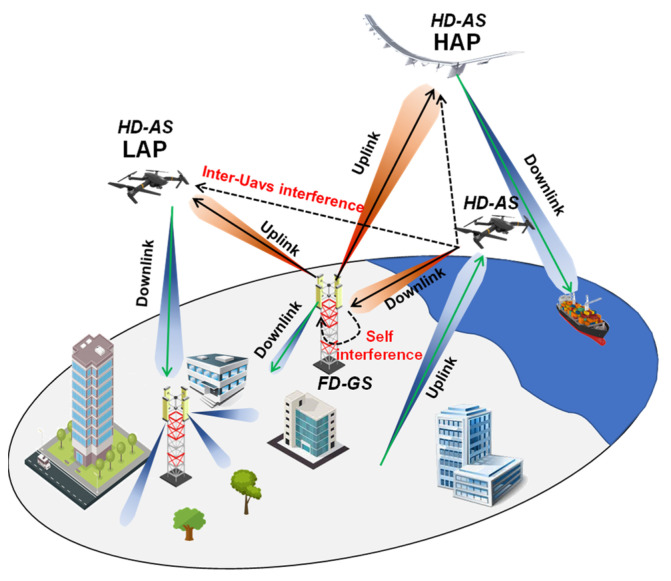
Example of Hybrid-Duplex (HBD) wireless communication.

**Figure 7 sensors-22-03136-f007:**
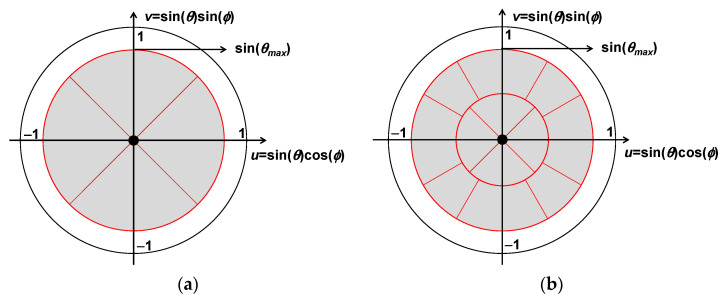
Example of cells configuration in the *uv* plane: (**a**) single layer with *n* = 8 cells and (**b**) two layers with *n* = 20 cells.

**Figure 8 sensors-22-03136-f008:**
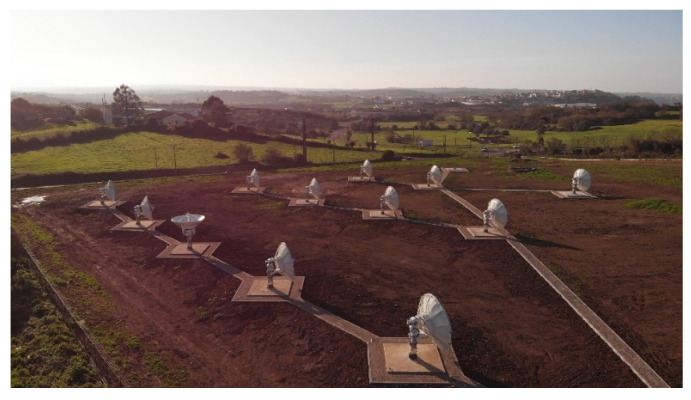
OneWeb SNP, Sintra, Portugal.

**Figure 9 sensors-22-03136-f009:**
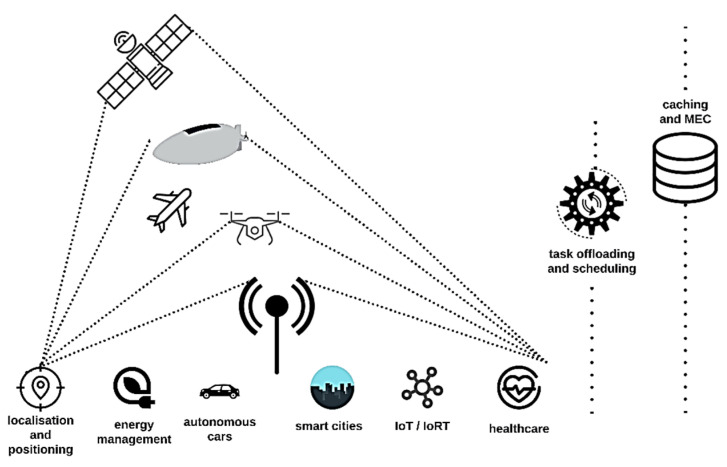
Common application scenarios by leveraging SAGINs and key services offered by the network infrastructure at different layers of the SAGIN.

**Table 1 sensors-22-03136-t001:** An overview of survey papers dealing with 6G and SAGIN.

Paper	Main Contribution	Focus on
Arum et al., 2020 [45]	Review of the role played by High-Altitude Platforms (HAPs) in exploiting cellular radio spectrum for wireless communications service in remote areas.	Overview on aerial platforms, network topology, coverage and hap-terrestrial joint exploitation. Mostly devoted to cellular networks.
Ye et al., 2020 [46]	SAGIN system from the perspective of cooperative communication point of view. The approximated and asymptotic closed-form expressions for outage probabilities of each link as well as the outage probability of the SAGIN system have been derived.	Cooperation between high-altitude platforms (HAPs) and terrestrial base stations (BSs) for serving communication from geostationary (GEO) satellites and the user. System outage performance are analyzed in detail. Mostly dealing with SAGIN.
Yaacoub et al., 2020 [47]	Thorough survey on fronthaul and backhaul technologies that offer the 6G connectivity in rural areas.	Mostly devoted to 6G
Guo et al., 2021 [48]	Overview of quasi-optical techniques employed in multi-beam antennas for B5G and 6G mmWave and THz networks.	Antennas for terrestrial and non-terrestrial wireless communications networks for Beyond 5G (B5G) and 6G with emphasis on mmWave and Terahertz frequency range.
Ray, 2021 [26]	Basics behind the SAGIN and 6G and their convergence into 6G-SAGIN, with particular attention devoted to the role of unmanned aerial vehicles (UAVs).	Enabling technologies for 6G, SAGIN and their synergic use. Research challenges and future directions on these topics.
Jiang et al., 2021 [49]	Survey on UAV communications for 6G and analysis of their energy consumption.	Mostly devoted to 6G.
Cheng et al., 2021 [50]	Service-oriented SAGINs management architecture.	Two categories of enabling key technologies, heterogeneous resource orchestration technologies and cloud-edge synergy technologies are addressed and discussed. Mostly devoted to 6G SAGIN.
Zhao et al., 2021 [51]	Overview of some promising technologies in 6G networks with focus on AI, intelligent surfaces, terahertz and cell-free massive MIMO.	Mostly devoted to 6G. Security and privacy techniques that can be applied to protect 6G data.
Wang et al., 2022 [52]	Survey of the integration of blockchain technologies for securing Space–Air–Ground Internet of Things (SAG-IoT) applications.	Analysis of architecture, characteristics, and security threats of SAG-IoT systems. Challenges in blockchain integration and artificial intelligence exploitation in the SAG-IoT framework.
Wei et al., 2022 [53]	SAGIN architecture exploitation for enabling Immersive Media (IM) services.	Architectural challenges for SAGIN in supporting low-latency and high reliability services.
This work	Review of technological solutions and advances in the framework of a Vertical Heterogeneous Network (VHetNet) integrating satellite, airborne and terrestrial networks.	Strong emphasis on the available antenna systems in satellite, airborne and ground layers. SAGIN and 6G are both considered. Overview on applications exploiting these frameworks.

**Table 2 sensors-22-03136-t002:** Classification of small satellites [59].

Classification	Mass	CubeSats and PocketQubes *	Industrial Developments and Products
FemtoSat	<0.1 kg		
PicoSat	0.1 to 1 kg	0.25U/1 to 3 p	SpaceBEE (Swarm Technologies), Unicorn-2 (Alba Orbital)
NanoSat	1 to 10 kg	1 to 6 U	Dove (Planet), LEMUR (Spire)
MicroSat	10 to 100 kg	8 to 27 U	8U, 12U, 16U platforms (GomSpace), up to 12U (EnduroSat), up to 27U (HEX20), VesselSat (LuxSpace)
MiniSat	100 to 500 kg		Starlink (SpaceX), OneWeb

* assuming a typical mass of less than 1.33 kg (3 lbs) per U and 250 g per p.

**Table 3 sensors-22-03136-t003:** Key parameters of typical satellite Earth orbits.

Orbit	Altitude	Onboard Angular Range	Visibility Time	Latency
VLEO	<500 km	Beyond ± 60°	<20 min.	<20 ms
LEO	~1000 km	±60°	20 min.	~20 ms
MEO	~10,000 km	±20°	45 min.	~100 ms
GEO	35,786 km	±8.7°	Permanent	~250 ms
HEO	Up to 40,000 km at apogee	±10°	A few hours	~250 ms

**Table 4 sensors-22-03136-t004:** Evolution of broadband satellite antenna systems.

Reference	Frequency Band	No. of Main Reflectors	Sub-Reflector	Configuration	Feed Systems
[78]	K/Ka-band	8 ^(1)^	--	SFB	Single-band dual-CP
[79,80]	K/Ka-band	4	--	SFB	Dual-band dual-CP
[81]	K/Ka-band	2	--	MFB	Single-band dual-CP up to 25 feeds per beam
[82]	K/Ka-band	2	--	MFB	Single-band dual-CP 7 feeds per beam
[83] ^(2)^	K/Ka-band	2	--	MFB	Dual-band dual-CP 4 feeds per beam
[84]	K/Ka-band	2	Gridded	SFB	Dual-band dual-LP
[85]	K/Ka-band	1	Dichroic	MFB	Single-band dual-CP 7 feeds per beam
[83] ^(3)^	K/Ka-band	1	--	MFB	Dual-band dual-CP 4 feeds per beam

^(1)^ Eight user link antennas plus two dedicated tracking antennas [78]. ^(2)^ Antenna solution described in Section III.A in [83]. ^(3)^ Antenna solution described in Section III.B in [83].

**Table 5 sensors-22-03136-t005:** SAGIN-related relevant scenarios and enabling technologies in the literature.

	Name	References	SAGIN Role and Relevance
**Application Scenarios and Industry Verticals**	Autonomous Driving	[50]	support terrestrial networks in meeting QoS level; on-demand resources and services to be deployed
	Smart City	[18,50,199]	provide coverage extension services, on-demand bandwidth, monitoring services, mobile crowdsensing; monitoring capabilities and fast deployment in fast-changing environments (such as cities)
	IoRT	[18,26,200,201]	remote IoT scenarios (connectivity, custom services); NTN nodes are the most viable option
	IoV	[202,203]	on-demand resources and services to be deployed; coverage in poorly connected areas
	Healthcare services	[26,200]	telemedicine and e-health services; coverage in poorly connected areas
	Maritime monitoring	[26,200]	life-saving support, deep sea exploration, under sea research activities, real-time command and control of autonomous ships; only viable connectivity option
	Energy distribution and monitoring	[200]	control of critical energy infrastructures; monitoring in remote/not covered areas
	Continuum indoor-outdoor localization and positioning	[204,205]	realization of an integrated indoor-outdoor localization and positioning system, working in the absence of GNSS capabilities or in urban canyons
**Enabling technologies and services**	Coverage extension	[50]	deploy and use of NTN nodes to provide (additional) coverage to high-traffic or uncovered areas to support user services or to complement/substitute the terrestrial infrastructure
	Mobility management	[26,197]	predict and control the 3D mobility of NTN nodes mobility to guarantee user QoS; orchestration and management to reduce link interruptions
	Task scheduling and offloading	[26,50,197,201,202,206,207,208]	offload task to NTN nodes to save local computational resources or to run too intensive tasks, scheduling them to respect QoS level; on-demand additional computational power to be deployed
	Mobile crowdsensing and MEC	[18,26,202,209]	mobile crowdsensing to safeguard the network from edge caching issues; UAVs as base stations for services to other UAVs or ground/satellite stations
	Caching and on-the-fly data center	[26,206,209]	NTN nodes providing caching capabilities to guarantee low delay; edge capabilities in combination with remote cloud support

## Data Availability

Not applicable.

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
