# Peer review of "Space-Air-Ground Integrated 6G Wireless Communication Networks: A Review of Antenna Technologies and Application Scenarios"

_sensors, 2022, doi:10.3390/s22093136_

Round 1
Reviewer 1 Report
This paper is well-written.
- There is not a numerical simulation to explain the performance of different schemes. Please consider to add.
- In the application scenarios, please use figures (such as Fig. 5 and 6) to illustrate different application scenarios.
- Please chech the typos.
- The delayed information induced by long propagation between ground and satellite is not discussed enough. In fact, for antennas, multiplexing gain, equivalently degrees-of-freedom, is a very important metric for spectral efficiency. I suggest the authors discuss the delayed feedback from degrees-of-freedom perspective. In particular, the following papers should be properly commented in this paper: "Achievable DoF Regions of Three-User MIMO Broadcast Channel with Delayed CSIT"
Author Response
Paper No.: sensors-1687937
Paper Title: Space-Air-Ground Integrated 6G Wireless Communication Networks: A review of Antenna Technologies and Application Scenarios
Authors: Francesco Alessio Dicandia, Nelson J. G. Fonseca, Manlio Bacco, Sara Mugnaini, Simone Genovesi
Reviewer: 1
The paper is well-written.
Rev: 1.1) There is not a numerical simulation to explain the performance of different schemes. Please consider to add.
Au: We would like to thank the reviewer for the useful remarks and observations that helped to improve the paper quality. Numerical simulations represent a useful tool for the design and optimization of any system or device. However, the submitted work is a theoretical investigation and a review of technological solutions and advances in the framework of a Vertical Heterogeneous Network (VHetNet) integrating satellite, airborne and terrestrial networks. Therefore, we think that simulations are out of the scope of the present paper since we are not proposing a particular design of a system or part of it but a comprehensive analysis of this kind of network and its components.
Rev: 1.2) In the application scenarios, please use figures (such as Fig. 5 and 6) to illustrate different application scenarios. (MANLIO)
Au: We would like to thank the reviewer for the valuable comment. A new figure has been added to the revised paper (Figure 9) to graphically depict the application scenarios covered in Section 5, as well as the key services offered – as described in the literature – by leveraging the different layers of a SAGIN and their capabilities. Moreover, the new figure is reported hereafter (Fig. R1):
|
Fig. R1. Common application scenarios by leveraging SAGINs and key services offered by the network infrastructure at different layers of the SAGIN. |
The following sentence has been added in the revised paper:
“Some common application scenarios by leveraging SAGINs and key services offered by the network infrastructure at different layers of the SAGIN are graphically illustrated in Figure 9.”
Rev: 1.3) Please check the typos.
Au: We thank the reviewer for the suggestion. We revised the paper.
Rev: 1.4) The delayed information induced by long propagation between ground and satellite is not discussed enough. In fact, for antennas, multiplexing gain, equivalently degrees-of-freedom, is a very important metric for spectral efficiency. I suggest the authors discuss the delayed feedback from degrees-of-freedom perspective. In particular, the following papers should be properly commented in this paper: "Achievable DoF Regions of Three-User MIMO Broadcast Channel with Delayed CSIT"
Au: We thank the reviewer for this suggestion. The question related to the delayed information has been added in the revised paper as well as the suggested reference.
In the revised paper the following sentences have been added in the sub-section 3.2:
“Indeed, a MIMO system with the corresponding MIMO channel matrix H, offers K=rank(HH*) parallel Single-Input-Single-Output (SISO) subchannels with different gain where it is possible to send different streams of data. In general, two scenarios are possible: absence of knowledge of the Channel State Information (CSI) at the transmitter and 2) knowledge of the CSI at the transmitter (CSIT). In the former case the transmitted power is evenly split to each Kth subchannel. Conversely, in case of CSIT, it is possible to obtain an optimum power allocation for each Kth subchannel (water filling technique) according to the eigenvalues (λi) of the MIMO channel matrix H in order to optimize the spectral efficiency. However, it is worth noting that, in practice, perfect CSIT is not possible due to errors on channel estimation and feedback delay [113]. MIMO performance in case of feedback delay and channel estimation errors have been analyzed in [114-116]. For the airborne layer, the CSIT concern turns out to be even more pronounced due to the high altitudes and the movements of LAPs and HAPs flying platforms [46]. An interesting solution for the CSIT mitigation has been proposed in [117] where a TB relay and an effective interference alignment scheme for maximizing the spectral efficiency of HAPs-ground stations communications were presented.”
Moreover, we have added the following references:
[46] Yaacoub, E.; Alouini, M.-S. A Key 6G Challenge and Opportunity—Connecting the Base of the Pyramid: A Survey on Rural Connectivity. Proc. IEEE 2020, 108, 533–582, doi:10.1109/JPROC.2020.2976703.
[113] Ramya, T.R.; Bhashyam, S. Using Delayed Feedback for Antenna Selection in MIMO Systems. IEEE Trans. Wirel. Commun. 2009, 8, 6059–6067, doi:10.1109/TWC.2009.12.090304.
[114] Zhang, T.; Wang, R. Achievable DoF Regions of Three-User MIMO Broadcast Channel With Delayed CSIT. IEEE Trans. Commun. 2021, 69, 2240–2253, doi:10.1109/TCOMM.2021.3049399.
[115] Vaze, C.S.; Varanasi, M.K. The Degrees of Freedom Region of the Two-User MIMO Broadcast Channel with Delayed CSIT. In Proceedings of the 2011 IEEE International Symposium on Information Theory Proceedings; IEEE: St. Petersburg, Russia, July 2011; pp. 199–203.
[116] Han, S.; Yang, C. Performance Analysis of MRT and Transmit Antenna Selection with Feedback Delay and Channel Estimation Error. In Proceedings of the 2007 IEEE Wireless Communications and Networking Conference; IEEE: Kowloon, China, 2007; pp. 1134–1138.
[117] Sudheesh, P.G.; Mozaffari, M.; Magarini, M.; Saad, W.; Muthuchidambaranathan, P. Sum-Rate Analysis for High Altitude Platform (HAP) Drones With Tethered Balloon Relay. IEEE Commun. Lett. 2018, 22, 1240–1243, doi:10.1109/LCOMM.2017.2785847.

Reviewer 2 Report
This paper provides a general overview concerning Space-Air-Ground Integrated Network (SAGIN) and emphasizes some research activities to support the multi-dimensional and inter-operational network of the future 6G wireless communications and beyond.
The paper idea is valid and added knowledge to the literature. The paper is related to the journal scope. Overall, the paper cannot be accepted and has to be improved.
The limitation of the paper lies in paper structures and its presentation as it has to be improved.
- In introduction section is missing to support the authors ideas.
- The related works aren't enough and a new recently published paper has to be cited. e.g:
- Ye, J., Dang, S., Shihada, B., & Alouini, M. S. (2020). Space-air-ground integrated networks: Outage performance analysis. IEEE Transactions on Wireless Communications, 19(12), 7897-7912.
- Veeraiah, N., Khalaf, O. I., Prasad, C. V. P. R., Alotaibi, Y., Alsufyani, A., Alghamdi, S. A., & Alsufyani, N. (2021). Trust aware secure energy efficient hybrid protocol for manet. IEEE Access, 9, 120996-121005.
- Zhao, Y., Zhai, W., Zhao, J., Zhang, T., Sun, S., Niyato, D., & Lam, K. Y. (2020). A comprehensive survey of 6g wireless communications. arXiv preprint arXiv:2101.03889.
- The SLR method is poor. The including and excluding criteria are missing.
- For table 2, it will be good idea if you can add 2 new columns for reference number.
- They have to add a new section before conclusion for the learn lessons or the recommends frameworks for 6G networking.
- The limitations and future research are missing.
- The paper has to be proofread.
Author Response
Paper No.: sensors-1687937
Paper Title: Space-Air-Ground Integrated 6G Wireless Communication Networks: A review of Antenna Technologies and Application Scenarios
Authors: Francesco Alessio Dicandia, Nelson J. G. Fonseca, Manlio Bacco, Sara Mugnaini, Simone Genovesi
Reviewer: 2
This paper provides a general overview concerning Space-Air-Ground Integrated Network (SAGIN) and emphasizes some research activities to support the multi-dimensional and inter-operational network of the future 6G wireless communications and beyond.
The paper idea is valid and added knowledge to the literature. The paper is related to the journal scope. Overall, the paper cannot be accepted and has to be improved.
The limitation of the paper lies in paper structures and its presentation as it has to be improved.
Rev: 2.1) In introduction section is missing to support the authors ideas
Au: We would like to thank the reviewer for the remark. The main paper contribution is to illustrate a comprehensive survey regarding recent advances and technical solutions in the design and development of breakthrough SAGINs for supporting seamless and ubiquitous wireless connectivity for future 6G wireless communications. Emphasis has been put on the available antenna systems in satellite, airborne and ground layers by highlighting challenges as well as by providing some interesting trends in research. Additionally, relevant application scenarios regarding the paradigm of SAGIN in present and future wireless communications are discussed, covering 5G, B5G and 6G use cases. These features have been emphasized in the submitted paper through different sentences. Some of these are reported hereafter:
Abstract:
“A review of technological solutions and advances in the framework of a Vertical Heterogeneous Network (VHetNet) integrating satellite, airborne and terrestrial networks is presented”
“Emphasis is put on the available antenna systems in satellite, airborne and ground layers by highlighting strengths and weakness as well as by providing some interesting trends in research. A summary of the most suitable applicative scenarios for future 6G wireless communications are finally illustrated.”
Introduction:
“This article provides a general overview concerning Space-Air-Ground Integrated Network (SAGIN) and emphasizes some research activities to support the multi-dimensional and inter-operational network of the future 6G wireless communications and beyond”
Conclusion:
“A comprehensive survey regarding recent advances and technical solutions in the design and development of breakthrough space-air-ground integrated networks for supporting seamless and ubiquitous wireless connectivity for future 6G wireless communications has been carried out.”
“Finally, relevant application scenarios regarding the paradigm of SAGIN in present and future wireless communications are discussed, covering 5G, B5G and 6G use cases. “
However, with the aim of improving the quality of the paper as well as to be clearer with the readers, a new sentence that further illustrate the paper scope has been added in the introduction of the revised paper and it is reported hereafter:
“Specifically, particular attention has been addressed on the available antenna systems in satellite, airborne and ground layers by highlighting strengths, challenges as well as by providing some promising research directions. Antennas are certainly among the most fundamental components, and they are determinant for the performance of the onboard transceiver subsystem.”
Moreover, to highlight the differences between the proposed review paper and the survey papers published in recent years dealing with 6G and SAGIN, a new table (Table 1 of the revised paper) has been introduced in the revised paper and reported hereafter:
Paper |
Main contribution focus |
this work |
Review of technological solutions and advances in the framework of a Vertical Heterogeneous Network (VHetNet) integrating satellite, airborne and terrestrial networks. Strong emphasis on the available antenna systems in satellite, airborne and ground layers. |
Ray, 2021 [26] |
Basics behind the SAGIN and 6G and their convergence into 6G-SAGIN, with particular attention devoted to the role of unmanned aerial vehicles (UAVs). |
Cheng et alii, 2021 [45] |
Service-oriented SAGINs management architecture. Two categories of enabling key technologies, heterogeneous resource orchestration technologies and cloud-edge synergy technologies are addressed and discussed. |
Yaacoub et alii, 2020 [46] |
Survey on fronthaul and backhaul technologies that offer the 6G connectivity in rural areas. |
Guo et alii, 2021 [47] |
Overview of quasi-optical techniques employed in multi-beam antennas for B5G and 6G mmWave and THz networks. |
Arum et alii, 2020 [48] |
Review of the role played by High-Altitude Platforms (HAPs) in exploiting cellular radio spectrum for wireless communications service in remote areas. |
Jiang et alii, 2021 [49] |
Survey on UAV communications for 6G and analysis of their energy consumption. |
Ye et alii, 2020 [50] |
SAGIN system from the perspective of cooperative communication point of view. The approximated and asymptotic closed-form expressions for outage probabilities of each link as well as the outage probability of the SAGIN system have been derived. |
Zhao at alii, 2021 [51] |
Overview of some promising technologies in 6G networks with focus on AI, intelligent surfaces, terahertz and cell-free massive MIMO. Discussion about security and privacy techniques that can be applied to protect 6G data. |
Rev: 2.2) The related works aren't enough and a new recently published paper has to be cited. e.g:
Ye, J., Dang, S., Shihada, B., & Alouini, M. S. (2020). Space-air-ground integrated networks: Outage performance analysis. IEEE Transactions on Wireless Communications, 19(12), 7897-7912.
Veeraiah, N., Khalaf, O. I., Prasad, C. V. P. R., Alotaibi, Y., Alsufyani, A., Alghamdi, S. A., & Alsufyani, N. (2021). Trust aware secure energy efficient hybrid protocol for manet. IEEE Access, 9, 120996-121005.
Zhao, Y., Zhai, W., Zhao, J., Zhang, T., Sun, S., Niyato, D., & Lam, K. Y. (2020). A comprehensive survey of 6g wireless communications. arXiv preprint arXiv:2101.03889.
Au: We thank the reviewer for the suggested references. Two of the three recommended references have been added on the revised paper.
Regarding the reference not included in the revised paper, namely:
- Veeraiah et al., ‘Trust Aware Secure Energy Efficient Hybrid Protocol for MANET’, IEEE Access, vol. 9, pp. 120996–121005, 2021, doi: 10.1109/ACCESS.2021.3108807.
we think that it is too far with respect to the proposed review paper topic. In fact, it deals with on Mobile Ad hoc Network (MANET) with focus on routing protocol. Moreover, the paper proposes a hybrid algorithm employing the Cat Salp Optimization (CSO) along with Salp Swarm Optimization (SSA) hybrid routing algorithms to decrease the energy loss during transmission and boost the duration of this system. Therefore, it has been decided not include it in the revised paper.
Rev: 2.3) The SLR method is poor. The including and excluding criteria are missing.
Au: The submitted paper presents an extensive overview about Space-Air-Ground Integrated 6G wireless communication networks with a major focus on antenna technologies and applicative scenarios. We believe that the paper provides a solid background of the idea at the basis of SAGIN, which is developed in its general structure, with a detailed focus on antenna systems technologies. Insights on open challenges and future directions are provided as well. The references are more than 200 and cover the most recent works on the different aspect of this topic. We therefore believe our paper can be of interest for readers that can want to enter this research field or to provide interesting pointers to researchers close to this subject.
Rev: 2.4) For table 2, it will be good idea if you can add 2 new columns for reference number.
Au: We thank the reviewer for the suggestion. However, Table 2 talks about general information that are somehow well-known. Therefore, we think that new references in Table 2 cannot improve the paper quality or make it more difficult to read. These information can be easily found interested authors.
Rev: 2.5) They have to add a new section before conclusion for the learn lessons or the recommends frameworks for 6G networking.
Au: We would like to thank the reviewer for the remark. As extensively described in the submitted paper, the main goal of the submitted paper is to provide a general overview about Space-Air-Ground Integrated Network (SAGIN) with a particular emphasis on antenna technologies in satellite, airborne and ground layers as well as a summary of the most suitable applicative scenarios about seamless and ubiquitous wireless connectivity in future 6G wireless communication networks. Therefore, recommendations about this topic are out of the scope of our work. The goal of the submitted paper review is not to make a sort of classification about “good” and “bad” antennas technologies or application scenarios. Conversely, it presents the most important technologies by highlighting strengths and challenges according to the context, namely space, air and ground segment, and it provides useful insight and key factors for future research trends.
Rev: 2.6) The limitations and future research are missing.
Au: We kindly disagree on this reviewer’s concern. Indeed, the paper deals with may important technological aspects with particular focus on antenna technologies by providing a thorough overview about the available solutions by explaining their limitations and strengths according to the scenario.
For instance, some important sentences about future research or limitations of the submitted paper are reported hereafter:
Space segment
“Further developments considered more advanced feed arrays with overlapping clusters in a multiple-feed-per-beam (MFB) configuration to reduce further the number of apertures to only 2 [79-81]”
“An alternative solution considered the use of a dichroic sub-reflector to produce a complete multiple beam coverage using a single large aperture [83]. This field of research is still very active as the renewal of existing broadcasting satellites provides an opportunity to embark secondary broadband payloads and the accommodation of the reflector antennas is always the main limiting factor.”
“Besides more conventional satellite payloads, there are several developments aiming at introducing higher flexibility through the use of reconfigurable phased array antennas, made possible thanks to major advances in the field of RFICs.”
“The solutions currently under development, including the OneSat programme of Airbus Defence and Space [85] and the INSPIRE programme of Thales Alenia Space [86] , aim at providing fully reconfigurable software defined payloads based on single-reflector imaging antenna geometries.”
Airborne segment
“Advances in 3D printed technology and manufacturing processes make Dielectric Resonator Antenna (DRA) technology another attractive solution for the development of commercial array antennas at mmWave [154].”
“With the purpose to reduce both cost and power consumption, key factors for future wireless communications, unconventional arrays designing such as sparse arrays [157,158], thinned arrays [159] and subarrays techniques [160,161] surely will represent a noteworthy airborne array designing technique in the future.”
“Another crucial aspect that phased array designers must face is the calibration [164,165].”
“However, it is worth observing that array calibration represents one of the main array costs and hence its usage must be accurately assessed by making a sort of tradeoff between the desired performance and overall cost [166]. For instance, within the framework of 5G, many phased arrays without the calibration procedure have been proposed [166-169] with the purpose to drastically reduce their cost by highlighting acceptable array performance degradation. Some phased array calibrations methods are described in [170,171].”
“Despite the advantageous of mmWave communications, such as larger spectrum, the adoption of large phased arrays for both UAVs and users mobility makes the antenna beam alignment between transmitter (TX) and receiver (RX) a challenging task to be tackled to guarantee the link robustness and hence satisfy the expected QoS [111]”
“In the mmWave and sub-terahertz domains, quasi-optical antenna solutions are also considered a promising alternative to reduce the number of control nodes while keeping high gain figures [47].”
Ground segment
“When mobility came along though, a traditional parabolic antenna was not the best option for all markets anymore as it is bulky, heavy, fits into an unappealing dome and not fulfilling the requirements of a terminal that need to be mounted on a possibly small moving, or flying, vehicle. The need for compact low-profile antennas for mobile terminals contributed to the development of more compact (and complex) geometries for steerable reflector antennas [178], which have the capability to fit into a smaller, more compact volume and to maintain contact with a GEO satellite while the vehicle is moving.”
“The continuous tracking adds a significant mechanical stress to the reflector antenna motors with respect to a traditional GSO user terminal. LEO tracking antennas also must move rapidly as a typical LEO satellite can stay in the visibility span of a user terminal (up to 120 degrees typically) as little as 10 seconds. This makes traditional reflector antennas not particularly suited for LEO applications.”
“Most flat panel systems also suffer regulatory issues as many existing regulatory requirements for Satcom user terminals are historically based on the parabolic-type antenna technology with invariant radiation patterns over antenna steering and more stringent side lobe level requirements. For the flat-panel type antenna, however, the radiation patterns are changing with beam steering, and it requires substantially more design efforts for a flat panel antenna to comply with a typical radiation mask.”
“In the most recent times, the most popular flat-panel antenna solutions for broadband satellite communication are surely phased arrays using either analog, digital or hybrid beamforming techniques. These antennas are also commercially known as Electronically Scanning Antennas (ESA).”
Rev: 2.7) The paper has to be proofread.
Au: We thank the reviewer for the comment. We revised the paper.

Reviewer 3 Report
Authors present a survey for integration of space air and ground in 6g. It is a good survey. However, there are some comments that should be considered to make the paper more comprehensive and readable.
1-authors should create table to identify the different between their work and previous work such as A review on 6G for space-air-ground integrated network: Key enablers, open challenges, and future direction, Machine learning for smart environments in B5G networks: connectivity and QoS,
6G service-oriented space-air-ground integrated network: A survey, Space-Air-Ground Integrated Network: A Survey and etc.
2-I suggest authors to make table to summarize the related work
In Airborne Segment:
3- authors missing to discuss tethered balloon technology for cover large area . which discuss and has several applications
4-Authors are missing discussing the Airborne Segment with details, so i suggest authors divide it into three sub sub sections i.e HAP , MAP,LAP.
6-I suggest authors to make table for each section to summarize the studeies such drones for smart cities and public safety, and QoS from HAP
7- Antenna radiation pattern needs to be discuss such as HAP antenna radiation pattern for providing coverage and service characteristics
Author Response
Paper No.: sensors-1687937
Paper Title: Space-Air-Ground Integrated 6G Wireless Communication Networks: A review of Antenna Technologies and Application Scenarios
Authors: Francesco Alessio Dicandia, Nelson J. G. Fonseca, Manlio Bacco, Sara Mugnaini, Simone Genovesi
Reviewer: 3
Authors present a survey for integration of space air and ground in 6g. It is a good survey. However, there are some comments that should be considered to make the paper more comprehensive and readable.
Rev: 3.1) Authors should create table to identify the different between their work and previous work such as A review on 6G for space-air-ground integrated network: Key enablers, open challenges, and future direction, Machine learning for smart environments in B5G networks: connectivity and QoS, 6G service-oriented space-air-ground integrated network: A survey, Space-Air-Ground Integrated Network: A Survey and etc.
Au: We would like to thank the reviewer for this question and the following one.
In order to provide an agile but clear comparison among the previous works on a similar subject, we have added a table (Table 1 of the revised paper) with review papers published in recent years dealing with 6G and space-air-ground integrated network (SAGIN), highlighting the particular angle from which they have looked to this broad topic.
Paper |
Main contribution focus |
this work |
Review of technological solutions and advances in the framework of a Vertical Heterogeneous Network (VHetNet) integrating satellite, airborne and terrestrial networks. Strong emphasis on the available antenna systems in satellite, airborne and ground layers. |
Ray, 2021 [26] |
Basics behind the SAGIN and 6G and their convergence into 6G-SAGIN, with particular attention devoted to the role of unmanned aerial vehicles (UAVs). |
Cheng et alii, 2021 [45] |
Service-oriented SAGINs management architecture. Two categories of enabling key technologies, heterogeneous resource orchestration technologies and cloud-edge synergy technologies are addressed and discussed. |
Yaacoub et alii, 2020 [46] |
Survey on fronthaul and backhaul technologies that offer the 6G connectivity in rural areas. |
Guo et alii, 2021 [47] |
Overview of quasi-optical techniques employed in multi-beam antennas for B5G and 6G mmWave and THz networks. |
Arum et alii, 2020 [48] |
Review of the role played by High-Altitude Platforms (HAPs) in exploiting cellular radio spectrum for wireless communications service in remote areas. |
Jiang et alii, 2021 [49] |
Survey on UAV communications for 6G and analysis of their energy consumption. |
Ye et alii, 2020 [50] |
SAGIN system from the perspective of cooperative communication point of view. The approximated and asymptotic closed-form expressions for outage probabilities of each link as well as the outage probability of the SAGIN system have been derived. |
Zhao at alii, 2021 [51] |
Overview of some promising technologies in 6G networks with focus on AI, intelligent surfaces, terahertz and cell-free massive MIMO. Discussion about security and privacy techniques that can be applied to protect 6G data. |
The following sentence has been added in the revised paper:
“Table 1 provides a comparison among the review papers published in recent years dealing with 6G and SAGIN technology, highlighting the particular angle from which they have looked to this broad topic.”
Rev: 3.2) I suggest authors to make table to summarize the related work
Au: We thank the reviewer for the suggested remark. We have included the table described in the previous answer.
Rev: 3.3) In Airborne Segment: authors missing to discuss tethered balloon technology for cover large area which discuss and has several applications.
Au: We would like to thank the reviewer for the suggestion. The tethered balloon technology has been added in the revised paper by introducing the following sentences:
“Tethered Balloons (TBs) represent another promising technology solution within the framework of LAP flying platforms for delivering wireless communications [104]. TBs consist of inflated balloon, usually filled with helium to lift in air ad hoc equipment, operating in the troposphere at an altitude of around 200 m – 500 m and are tethered to ground by several ropes [46]. These flying platforms have the potential to deliver a wide variety of wireless communication services by overcoming various challenges faced in terrestrial or satellite segment such as limited coverage area, delay and lack of Line of Sight (LoS) links [105]. Their usage has been exploited in different scenarios spanning from emergency communication systems, observation, military applications as well as antenna radiation pattern measurements [105-107]. Other examples where TBs have been also exploited deal with fronthaul WiFi access by using high gain omnidirectional antennas [108] as well as WiMAX connectivity [105]. Besides, in [109] TB has been employed for improving the achievable end-to-end data rate of ground users.
Moreover, the following references have been introduced:
[46] Yaacoub, E.; Alouini, M.-S. A Key 6G Challenge and Opportunity—Connecting the Base of the Pyramid: A Survey on Rural Connectivity. Proc. IEEE 2020, 108, 533–582, doi:10.1109/JPROC.2020.2976703.
[104] Alsamhi, S.H.; Almalki, F.A.; Ma, O.; Ansari, M.S.; Angelides, M.C. Performance Optimization of Tethered Balloon Technology for Public Safety and Emergency Communications. Telecommun. Syst. 2020, 75, 235–244, doi:10.1007/s11235-019-00580-w.
[105] Alsamhi, S.H.; Gupta, S.K.; Rajput, N.S. Performance Evaluation of Broadband Service Delivery via Tethered Balloon Technology. In Proceedings of the 2016 11th International Conference on Industrial and Information Systems (ICIIS); IEEE: Roorkee, India, December 2016; pp. 133–138.
[106] Saif, A.; Dimyati, K.; Noordin, K.A.; Shah, N.S.M.; Alsamhi, S.H.; Abdullah, Q. Energy-Efficient Tethered UAV Deployment in B5G for Smart Environments and Disaster Recovery. In Proceedings of the 2021 1st International Conference on Emerging Smart Technologies and Applications (eSmarTA); IEEE: Sana’a, Yemen, August 10 2021; pp. 1–5.
[107] Steele, J. Measurement of Antenna Radiation Patterns Using a Tethered Balloon. IEEE Trans. Antennas Propag. 1965, 13, 179–180, doi:10.1109/TAP.1965.1138381.
[108] Bilaye, P.; Gawande, V.N.; Desai, U.B.; Raina, A.A.; Pant, R.S. Low Cost Wireless Internet Access for Rural Areas Using Tethered Aerostats. In Proceedings of the 2008 IEEE Region 10 and the Third international Conference on Industrial and Information Systems; IEEE: Kharagpur, India, December 2008; pp. 1–5.
[109] Alzidaneen, A.; Alsharoa, A.; Alouini, M.-S. Resource and Placement Optimization for Multiple UAVs Using Backhaul Tethered Balloons. IEEE Wirel. Commun. Lett. 2020, 9, 543–547, doi:10.1109/LWC.2019.2961906.
Rev: 3.4) In Airborne Segment: Authors are missing discussing the Airborne Segment with details, so i suggest authors divide it into three sub sub sections i.e HAP , MAP,LAP.
Au: We would like to thank the reviewer for the idea. We have thought about similar ideas during the planning of our work, but we have had also to manage the overall length of the paper and avoid to trade readability for adding many other (interesting) details and analysis. Therefore, we decided to not go too much into further details about the airborne layer. Indeed, as remarked in the paper introduction, the aim of this work is to illustrate an extensive overview concerning Space-Air-Ground Integrated Network (SAGIN) with emphasis on the available antenna systems in satellite, airborne and ground layers by highlighting strengths, challenges, interesting trends in research as well as relevant application scenarios regarding the present and future wireless communications covering 5G, B5G and 6G use cases.
Rev: 3.5) In Airborne Segment: I suggest authors to make table for each section to summarize the studeies such drones for smart cities and public safety, and QoS from HAP .
Au: We would like to thank the reviewer for the idea. However, as replied to the previous question, we believe that the introduction of new information about this section could get too far from the paper scope.
Rev: 3.6) In Airborne Segment: Antenna radiation pattern needs to be discuss such as HAP antenna radiation pattern for providing coverage and service characteristics
Au: We thank the reviewer for the helpful suggestion. In the revised paper, the question related to coverage of flying platforms has been further addressed by introducing a new figure (Figure 7 of the revised paper) reported in Fig. R1.
Moreover, the following sentences have been added:
“Concerning the coverage area, the flying platforms scenario differs from that of ground segment, whose coverage is typically rectangular in the uv plane, since a circular scan area turns out to be more appropriate [89]. Then, within a predefined circular scenario there could be several possible cells configuration. Figure 7 illustrates two examples of cells configuration within a circular scanning area with a maximum coverage angle of sin(qmax) off broadside direction (q = 0°), namely with N = 8 cells (Figure 7a) and N = 20 cells (Figure 7b).”
Figure 7
“It is worth observing that the cell configuration scheme has a strong impact on the UAV antenna parameters such as the Half Power Beam Width (HPBW) on both horizontal and vertical planes as well as lateral lobes.”
|
|
(a) |
(b) |
Fig. R1. Example of cells configuration in the uv plane: (a) single layer with N = 8 cells and (b) two layers with N = 20 cells. |
Regarding some examples of HAPs antennas, we believe that the submitted paper presents a comprehensive overview. Some sentences about HAP antenna are reported hereafter:
“Some solutions regarding the design of radiating systems for HAPs are illustrated in [144-146].”
“A Ka-band phased array for HAPs application composed of open-ended substrate-integrated square waveguides and a 4-channel beamformer circuit produced by Anokiwave was described in [152]. A relevant mmWave beam steering 8x8 array design solution operating from 26.5 GHz to 31 GHz for 5G BSs based on gap waveguide technology is presented in [153].”

Round 2
Reviewer 1 Report
Good Revision No Comments
Author Response
Paper No.: sensors-1687937 (round2)
Paper Title: Space-Air-Ground Integrated 6G Wireless Communication Networks: A review of Antenna Technologies and Application Scenarios
Authors: Francesco Alessio Dicandia, Nelson J. G. Fonseca, Manlio Bacco, Sara Mugnaini, Simone Genovesi
Reviewer: 1
Good Revision No Comments.
Au: We are glad that our revision has properly addressed the reviewer’s concerns. We would like to thank the reviewer for the useful remarks and observations that helped to improve the paper quality.
Reviewer 2 Report
The paper has been improved. It can be accepted in current form.
Author Response
Paper No.: sensors-1687937 (round2)
Paper Title: Space-Air-Ground Integrated 6G Wireless Communication Networks: A review of Antenna Technologies and Application Scenarios
Authors: Francesco Alessio Dicandia, Nelson J. G. Fonseca, Manlio Bacco, Sara Mugnaini, Simone Genovesi
Reviewer: 2
The paper has been improved. It can be accepted in current form.
Au: We would like to thank the reviewer for the remark. We are glad that our revision has properly addressed the reviewer’s concerns.
Reviewer 3 Report
The authors addressed my comments very well and the paper improved. It may consider for minor revision now
my comments are
1- last two raw in table 1 without years
2- the comparison between previous studies and current one is based on highlighted only, I suggest authors to add more columns for comparison in order to show the different in very clear way. then i can say that paper has new contribution. for example, Authors focused on Antenna technology while others not survey SAGIN. authors may add the columns for 6g, Antenna and etc..
3- authors should shift the first raw in table 1 to the end
5- I did not see any paper in the table from 2022, so I suggest authors update table with more recent works such as AI-Based Cloud-Edge-Device Collaboration in 6G Space-Air-Ground Integrated Power IoT, NOMA-Based Energy-Efficiency Optimization for UAV Enabled Space-Air-Ground Integrated Relay Networks, UAV-Assisted RF/FSO Relay System for Space-Air-Ground Integrated Network: A Performance Analysis etc.
Author Response
Paper No.: sensors-1687937 (round2)
Paper Title: Space-Air-Ground Integrated 6G Wireless Communication Networks: A review of Antenna Technologies and Application Scenarios
Authors: Francesco Alessio Dicandia, Nelson J. G. Fonseca, Manlio Bacco, Sara Mugnaini, Simone Genovesi
Reviewer: 3
The authors addressed my comments very well and the paper improved. It may consider for minor revision now. My comments are:
Rev: 3.1) - last two raw in table 1 without years.
Au: We have amended this error.
Rev: 3.2) the comparison between previous studies and current one is based on highlighted only, I suggest authors to add more columns for comparison in order to show the different in very clear way. then I can say that paper has new contribution. for example, Authors focused on Antenna technology while others not survey SAGIN. authors may add the columns for 6g, Antenna and etc..
Au: We have revised Table 1 accordingly to your suggestion. The amended table is also reported hereafter:
Paper |
Main contribution |
Focus on |
Arum et alii, 2020 [45] |
Review of the role played by High-Altitude Platforms (HAPs) in exploiting cellular radio spectrum for wireless communications service in remote areas. |
Overview on aerial platforms, network topology, coverage and hap-terrestrial joint exploitation. Mostly devoted to cellular networks. |
Ye et alii, 2020 [46] |
SAGIN system from the perspective of cooperative communication point of view. The approximated and asymptotic closed-form expressions for outage probabilities of each link as well as the outage probability of the SAGIN system have been derived. |
Cooperation between high-altitude platforms (HAPs) and terrestrial base stations (BSs) for serving communication from geostationary (GEO) satellites and the user. System outage performance are analyzed in details. Mostly dealing with SAGIN. |
Yaacoub et alii, 2020 [47] |
Thorough survey on fronthaul and backhaul technologies that offer the 6G connectivity in rural areas. |
Mostly devoted to 6G |
Guo et alii, 2021 [48] |
Overview of quasi-optical techniques employed in multi-beam antennas for B5G and 6G mmWave and THz networks. |
Antennas for terrestrial and non-terrestrial wireless communications networks for Beyond 5G (B5G) and 6G with emphasis on mmWave and Terahertz frequency range. |
Ray, 2021 [26] |
Basics behind the SAGIN and 6G and their convergence into 6G-SAGIN, with particular attention devoted to the role of unmanned aerial vehicles (UAVs). |
Enabling technologies for 6G, SAGIN and their synergic use. Research challenges and future directions on these topics. |
Jiang et alii, 2021 [49] |
Survey on UAV communications for 6G and analysis of their energy consumption. |
Mostly devoted to 6G. |
Cheng et alii, 2021 [50] |
Service-oriented SAGINs management architecture. |
Two categories of enabling key technologies, heterogeneous resource orchestration technologies and cloud-edge synergy technologies are addressed and discussed. Mostly devoted to 6G SAGIN. |
Zhao at alii, 2021 [51] |
Overview of some promising technologies in 6G networks with focus on AI, intelligent surfaces, terahertz and cell-free massive MIMO. |
Mostly devoted to 6G. Security and privacy techniques that can be applied to protect 6G data. |
Wang at alii, 2022 [52] |
Survey of the integration of blockchain technologies for securing Space Air Groung-Internet of Things (SAG-IoT) applications. |
Analysis of architecture, characteristics, and security threats of SAG-IoT systems. Challenges in blockchain integration and artificial intelligence exploitation in the SAG-IoT framework. |
Wei et alii, 2022 [53] |
SAGIN architecture exploitation for enabling Immersive Media (IM) services. |
Architectural challenges for SAGIN in supporting low-latency and high reliability services. |
This work |
Review of technological solutions and advances in the framework of a Vertical Heterogeneous Network (VHetNet) integrating satellite, airborne and terrestrial networks. |
Strong emphasis on the available antenna systems in satellite, airborne and ground layers. SAGIN and 6G are both considered. Overview on applications exploiting these frameworks. |
Rev: 3.3) authors should shift the first raw in table 1 to the end
Au: We have followed your suggestion and put all the references in chronological order.
Rev: 3.4) - I did not see any paper in the table from 2022, so I suggest authors update table with more recent works such as AI-Based Cloud-Edge-Device Collaboration in 6G Space-Air-Ground Integrated Power IoT, NOMA-Based Energy-Efficiency Optimization for UAV Enabled Space-Air-Ground Integrated Relay Networks, UAV-Assisted RF/FSO Relay System for Space-Air-Ground Integrated Network: A Performance Analysis etc.
Au: We have followed your advise and included some review papers from 2022. Specifically, in the revised paper it has been added reference [52] and [53].
[52] Wang, Y.; Su, Z.; Ni, J.; Zhang, N.; Shen, X. Blockchain-Empowered Space-Air-Ground Integrated Networks: Opportunities, Challenges, and Solutions. IEEE Commun. Surv. Tutor. 2022, 24, 160–209, doi:10.1109/COMST.2021.3131711.
[53] Wei, L.; Shuai, J.; Liu, Y.; Wang, Y.; Zhang, L. Service Customized Space-Air-Ground Integrated Network for Immersive Media: Architecture, Key Technologies, and Prospects. China Commun. 2022, 19, 1–13, doi:10.23919/JCC.2022.01.001.
